# FEDERATED VIRTUAL LEARNING ON HETEROGENEOUS DATA WITH LOCAL-GLOBAL DISTILLATION

## ABSTRACT

Dataset distillation is an emerging technique that generates virtual datasets with reduced size by keeping only the important information needed for a particular task. It has been shown that training model on distilled virtual datasets can provide higher training efficiency and alleviate the problem of inference attack in centralized learning. In this work, we investigate the utilization of distilled virtual data in federated learning (FL), dubbed as federated virtual learning. Although it is promising in enhancing synchronization, efficiency, and privacy in FL, *we discover that using distilled local datasets can amplify the heterogeneity issue.* To address this, we propose a new method, called **Fed**erated Virtual Learning on Heterogeneous Data with **L**ocal-**G**lobal **D**istillation (FEDLGD), which trains FL using *virtual data* created through a combination of local and global dataset distillation. Specifically, to harmonize the domain shifts, we propose iterative distribution matching to inpaint global information to *local virtual data* and use federated gradient matching to distill *global virtual data* that are shared with clients without compromising data privacy. By doing so, we can improve the consistency of local training by enforcing similarity between local and global features. We experiment on both benchmark and real-world datasets that contain heterogeneous data from different sources, and further scale up to an FL scenario that contains large number of clients with heterogeneous and class imbalance data. Our method outperforms *state-of-the-art* heterogeneous FL algorithms under various settings.

## 1 INTRODUCTION

In machine learning, synthetic data becomes more popular for supplementing or replacing real data when the latter is not suitable for direct use in training. Recently, dataset distillation (Wang et al., 2018; Cazenavette et al., 2022; Zhao et al., 2021; Zhao & Bilen, 2021; 2023), a technique that creates a smaller synthetic dataset while retaining similar model performance of that trained on the original dataset, has been explored in order to improve the efficiency and privacy of machine learning (Zhao et al., 2021; Zhao & Bilen, 2023; Dong et al., 2022). It can be naturally incorporated into Federated Learning (FL) (McMahan et al., 2017) to solve its problems when deployed in real world. For example, clients with different amounts of data cause asynchronization and affect the efficiency of FL systems. Dataset distillation addresses the issue by summarizing the same amount of synthetic datasets from the private local datasets for each client. We refer this effective strategy as *federated virtual learning*, as the models are trained from synthetic data (also referred as **virtual data**) (Xiong et al., 2022; Goetz & Tewari, 2020; Hu et al., 2022). These methods have been found to perform better than model-synchronization-based FL approaches while requiring fewer server-client interactions.

However, due to different data collection protocols, data from different clients inevitably face heterogeneity problems with domain shift, which means data may not be independent and identically distributed (iid) among clients. Heterogeneous data distribution among clients becomes a key challenge in FL, as aggregating model parameters from non-iid feature distributions suffers from client drift (Karimireddy et al., 2020) and diverges the global model update(Li et al., 2020b). Worse yet, we observe that using data distillation to synthesize local virtual data can amplify the heterogeneity issue. We empirically show in Fig. 1 with the tSNE plots of two different datasets, USPS (Netzer et al., 2011) and SynthDigits (Ganin & Lempitsky, 2015), each considered as a client. tSNE takes the original and distilled virtual images as input and embeds them into 2D planes. One can observe that the distribution becomes diverse after distillation. Quantitatively, we found that using dataset

distillation can amplify the statistical distances between the two datasets, with Wasswestein Distance and Maximum Mean Discrepancy (MMD) (Gretton et al., 2012) both increase [by] around 40%.

To alleviate the problem of data heterogeneity in classical FL settings, two main orthogonal approaches can be taken. *Approach 1* aims to minimize the difference between the local and global model parameters to improve convergence (Li et al., 2020a; Karimireddy et al., 2020; Wang et al., 2020). *Approach 2* enforces consistency in local embedded features using anchors and regularization loss (Tang et al., 2022; Zhou et al., 2022; Ye et al., 2022). The first approach can be easily applied to distilled local datasets, while the second approach has limitations when adapting to federated virtual learning. Specifically, VHL (Tang et al., 2022) samples global anchors from untrained StyleGAN (Karras et al., 2019) suffers performance drop when handling amplified heterogeneity after dataset distillation.

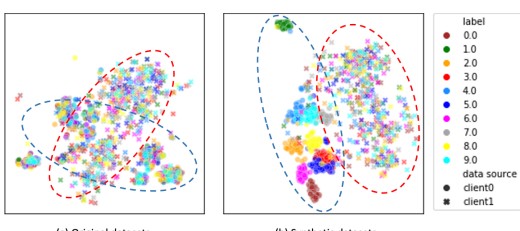

(a) Original datasets   (b) Synthetic datasets

Figure 1: Distilled local datasets can worsen heterogeneity in FL. tSNE plots of (a) original datasets and (b) distilled virtual datasets of USPS (client 0) and SynthDigits (client 1). The two distributions are marked in red and blue. We observe fewer overlapped ○ and × in (b) compared with (a), indicating higher heterogeneity between two clients after distillation.

Other methods, such as those that rely on external global data (Zhou et al., 2022), or feature sharing from clients (Ye et al., 2022), are less practical, as they pose greater data privacy risks compared to classical FL settings[1]. *Without hindering data privacy,* developing strategies following *approach 2* for federated virtual learning on heterogeneous data remains open questions on *1) how to set up global anchors for locally distilled datasets and 2) how to select the proper regularization loss(es).*

To this end, we propose FEDLGD, a federated virtual learning method with local and global distillation. We propose *iterative distribution matching* in local distillation by comparing the feature distribution of real and synthetic data using an evolving feature extractor. The local distillation results in smaller sets with balanced class distributions, achieving efficiency and synchronization while avoiding class imbalance. FEDLGD updates the local model on local distilled synthetic datasets (named *local virtual data*). We found that training FL with local virtual data can exacerbate heterogeneity in feature space if clients' data has domain shift (Fig. 1). Therefore, unlike previously proposed federated virtual learning methods that rely solely on local distillation (Goetz & Tewari, 2020; Xiong et al., 2022; Hu et al., 2022), we also propose a novel and efficient method, *federated gradient matching*, that integrated well with FL to distill global virtual data as anchors on the server side. This approach aims to alleviate domain shifts among clients by promoting similarity between local and global features. Note that we only share local model parameters w.r.t. virtual data. Thus, the privacy of local original data is preserved. We conclude our contributions as follows: 1) This paper focuses on an important but underexplored FL setting in which local models are trained on small virtual datasets, which we refer to as *federated virtual learning*. We design two effective and efficient dataset distillation methods for FL. 2) We are *the first* to reveal that when datasets are distilled from clients' data with domain shift, the heterogeneity problem can be *exacerbated* in the federated virtual learning setting. 3) We propose to address the heterogeneity problem by mapping clients to similar features regularized by gradually updated global virtual data using averaged client gradients. 4) Through comprehensive experiments on benchmark and real-world datasets, we show that FEDLGD outperforms existing state-of-the-art FL algorithms.

## 2 RELATED WORK

### 2.1 DATASET DISTILLATION

Data distillation aims to improve data efficiency by distilling the most essential feature in a large-scale dataset (e.g., datasets comprising billions of data points) into a certain terse and high-fidelity dataset. For example, Gradient Matching (Zhao et al., 2021) is proposed to make the deep neural network produce similar gradients for both the terse synthetic images and the original large-scale dataset. Besides, (Cazenavette et al., 2022) proposes matching the model training trajectory between real and

---

[1]Note that FedFA (Zhou et al., 2022), and FedFM (Ye et al., 2022) are unpublished works proposed concurrently with our work

synthetic data to guide the update for distillation. Another popular way of conducting data distillation is through Distribution Matching (Zhao & Bilen, 2023). This strategy instead, attempts to match the distribution of the smaller synthetic dataset with the original large-scale dataset. It significantly improves the distillation efficiency. Moreover, recent studies have justified that data distillation also preserves privacy (Dong et al., 2022; Carlini et al., 2022b), which is critical in federated learning. In practice, dataset distillation is used in healthcare for medical data sharing for privacy protection (Li et al., 2022). We refer the readers to (Sachdeva & McAuley, 2023) for other data distillation strategies.

## 2.2 HETEROGENEOUS FEDERATED LEARNING

FL performance downgrading on non-iid data is a critical challenge. A variety of FL algorithms have been proposed ranging from global aggregation to local optimization to handle this heterogeneous issue. *Global aggregation* improves the global model exchange process for better unitizing the updated client models to create a powerful server model. FedNova (Wang et al., 2020) notices an imbalance among different local models caused by different levels of training stage (e.g., certain clients train more epochs than others) and tackles such imbalance by normalizing and scaling the local updates accordingly. Meanwhile, FedAvgM (Hsu et al., 2019) applies the momentum to server model aggregation to stabilize the optimization. [Furthermore, there are strategies to refine the server model or client models using knowledge distillation such as FedDF (Lin et al., 2020), FedGen (Zhu et al., 2021b), FedFTG (Zhang et al., 2022), FedICT Wu et al. (2023), FedGKT (He et al., 2020), and FedDKC Wu et al. (2022). However, we consider knowledge distillation and data distillation two orthogonal directions to solve data heterogeneity issues.] *Local training optimization* aims to explore the local objective to tackle the non-iid issue in FL system. FedProx (Li et al., 2020a) straightly adds $L_2$ norm to regularize the client model and previous server model. Scaffold (Karimireddy et al., 2020) adds the variance reduction term to mitigate the "clients-drift". Also, MOON (Li et al., 2021b) brings mode-level contrastive learning to maximize the similarity between model representations to stable the local training. There is another line of works (Ye et al., 2022; Tang et al., 2022) proposed to use a global *anchor* to regularize local training. Global anchor can be either a set of virtual global data or global virtual representations in feature space. However, in (Tang et al., 2022), the empirical global anchor selection may not be suitable for data from every distribution as they don't update the anchor according to the training datasets.

## 2.3 DATASETS DISTILLATION FOR FL

Dataset distillation for FL is an emerging topic that has attracted attention due to its benefit for efficient FL systems. It trains model on distilled synthetic datasets, thus we refer it as federated virtual learning. It can help with FL synchronization and improve training efficiency by condensing every client's data into a small set. To the best of our knowledge, there are few published works on distillation in FL. Concurrently with our work, some studies (Goetz & Tewari, 2020; Xiong et al., 2022; Hu et al., 2022) distill datasets locally and share the virtual datasets with other clients/servers. Although privacy is protected against *currently* existing attack models, we consider sharing local virtual data a dangerous move. Furthermore, none of the existing work has addressed the heterogeneity issue.

## 3 METHOD

### 3.1 SETUP FOR FEDERATED VIRTUAL LEARNING

We start with describing the classical FL setting. Suppose there are $N$ parties who own local datasets $(D_1, \ldots, D_N)$, and the goal of a classical FL system, such as FedAvg (McMahan et al., 2017), is to train a global model with parameters $\theta$ on the distributed datasets ($D \equiv \bigcup_{i \in [N]} D_i$). The objective function is written as: $\mathcal{L}(\theta) = \sum_{i=1}^{N} |D_i|/|D|$, where $\mathcal{L}_i(\theta)$ is the empirical loss of client $i$. In practice, different clients in FL may have variant amounts of training samples, leading to asynchronized updates. In this work, we focus on a new type of FL training method – federated virtual learning, that trains on virtual datasets for efficiency and synchronization (discussed in Sec.2.3.) Federated virtual learning synthesizes local virtual data $\tilde{D}_i$ for client $i$ for $i \in [N]$ and form $\tilde{D} \equiv \bigcup_{i \in [N]} \tilde{D}_i$. Typically, $|\tilde{D}_i| \ll |D_i|$ and $|\tilde{D}_i| = |\tilde{D}_j|$. A basic setup for federated virtual learning is to replace $D_i$ with $\tilde{D}_i$ to teain FL model on the virtual datasets.

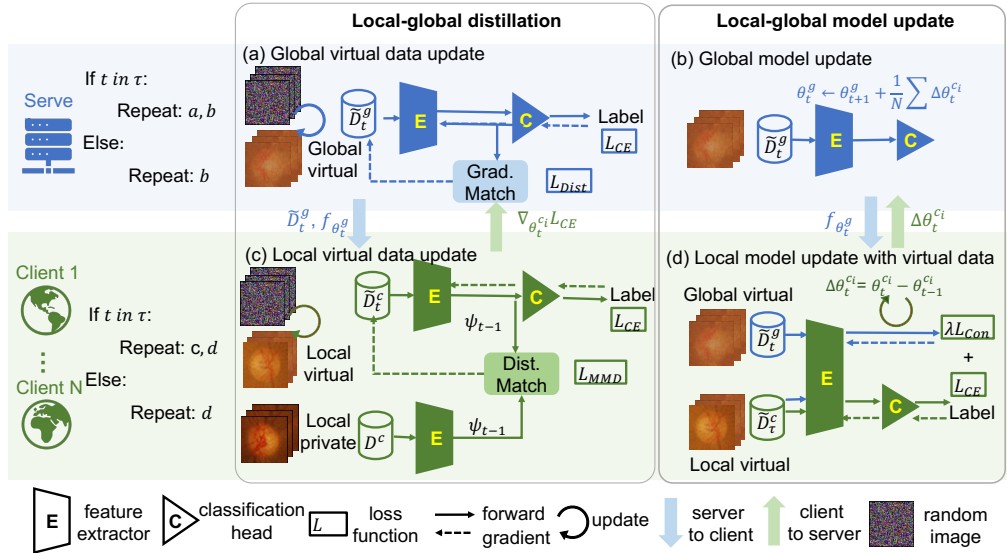

Figure 2: Overview for FEDLGD. We assume $T$ FL rounds will be performed, among which we will define the selected distillation rounds as $\tau \subset [T]$ for local-global iteration. For selected rounds ($t \in \tau$), clients will update local models (d) and refine the local virtual data with the latest network parameters (c), while the server uses aggregated gradients from cross-entropy loss ($\mathcal{L}_{\text{CE}}$) to update global virtual data (a) and update the global model (b). We term this procedure Iterative Local-global Distillation. For the unselected rounds ($t \in T \backslash \tau$), we perform ordinary FL pipeline on local virtual data with regularization loss ($\mathcal{L}_{\text{Con}}$) on global virtual data.

## 3.2 OVERALL PIPELINE

The overall pipeline of our proposed method contains three phases, including *1) initialization, 2) iterative local-global distillation, and 3) federated virtual learning.* We depict the overview of FEDLGD pipeline in Fig. 2. However, FL is inevitability affected by several challenges, including synchronization, efficiency, privacy, and heterogeneity. Specifically, we outline FEDLGD as follows:

We begin with the initialization of the clients' local virtual data $\tilde{D}^c$ by performing initial rounds of distribution matching (DM) (Zhao & Bilen, 2023). Meanwhile, the server will initialize global virtual data $\tilde{D}^g$ and network parameters $\theta_0^g$. In this stage, we generate the same amount of class-balanced virtual data for each client and server.

Then, we will refine our local and global virtual data using our proposed *local-global* distillation strategies in Sec. 3.3.1 and 3.3.2. This step is performed for a few selected iterations (e.g. $\tau = \{0, 5, 10\}$) to update $\theta$ using $\mathcal{L}_{\text{CE}}$ (Eq. 2), $\tilde{D}^g$ using $\mathcal{L}_{\text{Dist}}$ (Eq. 5), and $\tilde{D}^c$ using $\mathcal{L}_{\text{MMD}}$ (Eq. 1) in early training epochs. For each selected iterations, the server and clients will update their virtual data for a few distillation steps. For the unselected iterations, we perform training on $\theta$ using local virtual data $\tilde{D}^c$ with $\mathcal{L}_{\text{total}}$ (Eq. 2) which uses $\tilde{D}^g$ as regularization anchor to calculate $\mathcal{L}_{\text{Con}}$ (Eq. 4). We provide *implementation details, an algorithm box, theoretical analysis, more ablation studies and an anonymous link to our code* in the Appendix.

## 3.3 FL WITH LOCAL-GLOBAL DATASET DISTILLATION

### 3.3.1 LOCAL DATA DISTILLATION FOR FEDERATED VIRTUAL LEARNING

Our purpose is to decrease the number of local data to achieve efficient training to meet the following goals. First of all, we hope to synthesize virtual data conditional on class labels to achieve class-balanced virtual datasets. Second, we hope to distill local data that is best suited for the classification task. Last but not least, the process should be efficient due to the limited computational resource locally. To this end, we design Iterative Distribution Matching to fulfill our purpose.

**Iterative distribution matching.** We aim to gradually improve distillation quality during FL training. To begin with, we split a model into two parts, feature extractor $\psi$ (shown as $E$ in Fig. 2) and classification head $h$ (shown as $C$ in Fig. 2). The whole classification model is defined as $f^\theta = h \circ \psi$.

The high-level idea of distribution matching can be described as follows. Given a feature extractor $\psi : \mathbb{R}^d \to \mathbb{R}^{d'}$, we want to generate $\tilde{D}$ so that $P_\psi(D) \approx P_\psi(\tilde{D})$ where $P$ is the distribution in feature space. To distill local data during FL efficiently that best fits our task, we intend to use the up-to-date server model's feature extractor as our kernel function to distill better virtual data. Since we can't obtain ground truth distribution of local data, we utilize empirical maximum mean discrepancy (MMD) (Gretton et al., 2012) as our loss function for local virtual distillation:

$$\mathcal{L}_{\text{MMD}} = \sum_k^K || \frac{1}{|D_k^c|} \sum_{i=1}^{|D_k^c|} \psi^t(x_i) - \frac{1}{|\tilde{D}_k^{c,t}|} \sum_{j=1}^{|\tilde{D}_k^{c,t}|} \psi^t(\tilde{x}_j^t) ||^2, \tag{1}$$

where $\psi^t$ and $\tilde{D}^{c,t}$ are the server feature extractor and local virtual data from the latest global iteration $t$. Following (Zhao et al., 2021; Zhao & Bilen, 2023), we apply the differentiable Siamese augmentation on virtual data $\tilde{D}^c$. $K$ is the total number of classes, and we sum over MMD loss calculated per class $k \in [K]$. In such a way, we can generate balanced local virtual data by optimizing the same number of virtual data per class.

Although such an efficient distillation strategy is inspired by DM (Zhao & Bilen, 2023), we highlight the key difference that DM uses randomly initialized deep neural networks to extract features, whereas we use trained FL models with task-specific supervised loss. We believe *iterative updating* on the clients' data using the up-to-date network parameters can generate better task-specific local virtual data. Our intuition comes from the recent success of the empirical neural tangent kernel for data distribution learning and matching (Mohamadi & Sutherland, 2022; Franceschi et al., 2022). Especially, the feature extractor of the model trained with FEDLGD could obtain feature information from other clients, which further harmonize the domain shift between clients. We apply DM (Zhao & Bilen, 2023) to the baseline FL methods and demonstrate the effectiveness of our proposed iterative strategy in Sec. 4. Furthermore, note that FEDLGD only requires a few hundreds of local distillations steps using the local model's feature distribution, which is more computationally efficient than other bi-level dataset distillation methods (Zhao et al., 2021; Cazenavette et al., 2022).

**Harmonizing local heterogeneity with global anchors.** Data collected in different sites may have different distributions due to different collecting protocols and populations. Such heterogeneity will degrade the performance of FL. Worse yet, we found increased data heterogeneity among clients when federatively training with distilled local virtual data (see Fig. 1). We aim to alleviate the dataset shift by adding a regularization term in feature space to our total loss function for local model updating, which is inspired by (Tang et al., 2022; Khosla et al., 2020):

$$\mathcal{L}_{\text{total}} = \mathcal{L}_{\text{CE}}(\tilde{D}^g, \tilde{D}^c; \theta) + \lambda \mathcal{L}_{\text{Con}}(\tilde{D}^g, \tilde{D}^c), \tag{2}$$

$$\mathcal{L}_{\text{CE}} = \frac{1}{|\tilde{D}|} \sum_{x,y \in \tilde{D}} - \sum_k^K y_k log(\hat{y}_k), \hat{y} = f(x; \theta), \tag{3}$$

$$d(\psi_i; D^{Syn}, D_i) = \sum_{j \in B} - \frac{1}{|B_{\backslash j}^{y_j}|} \sum_{x_p \in B_{\backslash j}^{y_j}} \log \frac{\exp(\psi_i(x_j) \cdot \psi_i(x_p)/\tau_{temp})}{\sum_{x_a \in B_{\backslash j}} \exp(\psi_i(x_j) \cdot \psi_i(x_a)/\tau_{temp})} \tag{4}$$

$\mathcal{L}_{\text{CE}}$ is the cross-entropy measured on the virtual data $\tilde{D} = \{\tilde{D}^c, \tilde{D}^g\}$ and K is the number of classes. $\mathcal{L}_{\text{Con}}$ is the supervised contrastive loss where $B_{\backslash j}$ represents a batch containing both $\tilde{D}^c$ and $\tilde{D}^g$ but without data $j$, $B_{\backslash j}^{y_j}$ is a subset of $B_{\backslash j}$ only with samples belonging to class $y_j$, and $\tau_{temp}$ is a scalar temperature parameter. In such a way, global virtual data can be served for calibration, where $z_g$ is from $\tilde{D}^g$ as an anchor, and $z_p$ and $z_a$ are from $\tilde{D}^c$. At this point, a critical problem arises: ***What global virtual data shall we use?***

### 3.3.2    GLOBAL DATA DISTILLATION FOR HETEROGENEITY HARMONIZATION

Here, we provide an affirmative solution to the question of generating global virtual data that can be naturally incorporated into FL pipeline. Although distribution-based matching is efficient, local clients may not share their features due to privacy concerns. Therefore, we propose to leverage local clients' averaged gradients to distill global virtual data and utilize it in Eq. 4. We term our global data distillation method as *Federated Gradient Matching*.

Table 1: Test accuracy for `DIGITS` under different images per class (IPC) and model architectures. R and C stand for ResNet18 and ConvNet, respectively, and we set IPC to 10 and 50. There are five clients (MNIST, SVHN, USPS, SynthDigits, and MNIST-M) containing data from different domains. 'Average' is the unweighted test accuracy average of all the clients. The best performance under different models is highlighted using **bold**. The best results on ConvNet are marked in red and in black for ResNet18.

| `DIGITS` | | MNIST | | SVHN | | USPS | | SynthDigits | | MNIST-M | | Average | |
|---|---|---|---|---|---|---|---|---|---|---|---|---|---|
| IPC | | 10 | 50 | 10 | 50 | 10 | 50 | 10 | 50 | 10 | 50 | 10 | 50 |
| FedAvg | R | 73.0 | 92.5 | 20.5 | 48.9 | 83.0 | 89.7 | 13.6 | 28.0 | 37.8 | 72.3 | 45.6 | 66.3 |
| | C | 94.0 | 96.1 | 65.9 | 71.7 | 91.0 | 92.9 | 55.5 | 69.1 | 73.2 | 83.3 | 75.9 | 82.6 |
| FedProx | R | 72.6 | 92.5 | 19.7 | 48.4 | 81.5 | 90.1 | 13.2 | 27.9 | 37.3 | 67.9 | 44.8 | 65.3 |
| | C | 93.9 | 96.1 | 66.0 | 71.5 | 90.9 | 92.9 | 55.4 | 69.0 | 73.7 | 83.3 | 76.0 | 82.5 |
| FedNova | R | 75.5 | 92.3 | 17.3 | 50.6 | 80.3 | 90.1 | 11.4 | 30.5 | 38.3 | 67.9 | 44.6 | 66.3 |
| | C | 94.2 | 96.2 | 65.5 | 73.1 | 90.6 | 93.0 | 56.2 | 69.1 | 74.6 | 83.7 | 76.2 | 83.0 |
| Scaffold | R | 75.8 | 93.4 | 16.4 | 53.8 | 79.3 | 91.3 | 11.2 | 34.2 | 38.3 | 70.8 | 44.2 | 68.7 |
| | C | 94.1 | 96.3 | 64.9 | 73.3 | 90.6 | 93.4 | 56.0 | 70.1 | 74.6 | 84.7 | 76.0 | 83.6 |
| MOON | R | 15.5 | 80.4 | 15.9 | 14.2 | 25.0 | 82.4 | 10.0 | 11.5 | 11.0 | 35.4 | 15.5 | 44.8 |
| | C | 85.0 | 95.5 | 49.2 | 70.5 | 83.4 | 92.0 | 31.5 | 67.2 | 56.9 | 82.3 | 61.2 | 81.5 |
| VHL | R | 87.8 | 95.9 | 29.5 | 67.0 | 88.0 | 93.5 | 18.2 | 60.7 | 52.2 | **85.7** | 55.1 | 80.5 |
| | C | 95.0 | 96.9 | **68.6** | 75.2 | 92.2 | 94.4 | 60.7 | 72.3 | 76.1 | 83.7 | 78.5 | 84.5 |
| FEDLGD | R | **92.9** | **96.7** | **46.9** | **73.3** | **89.1** | **93.9** | **27.9** | **72.9** | **70.8** | 85.2 | **65.5** | **84.4** |
| | C | **95.8** | **97.1** | 68.2 | **77.3** | **92.4** | **94.6** | **67.4** | **78.5** | **79.4** | **86.1** | **80.6** | **86.7** |

**Federated gradient matching.** The concept of gradient-based dataset distillation is to minimize the distance between gradients from model parameters trained by original data and virtual data. It is usually considered as a learning-to-learn problem because the procedure consists of model updates and virtual data updates. Zhao *et al.* (Zhao et al., 2021) studies gradient matching in the centralized setting via bi-level optimization that iteratively optimizes the virtual data and model parameters. However, the implementation in (Zhao et al., 2021) is not appropriate for our specific context because there are two fundamental differences in our settings: 1) for model updating, the virtual dataset is on the server and will not directly optimize the targeted task; 2) for virtual data update, the 'optimal' model comes from the optimized local model aggregation. These two steps can naturally be embedded in local model updating and global virtual data distillation from the aggregated local gradients. First, we utilize the distance loss $\mathcal{L}_{Dist}$ (Zhao et al., 2021) for gradient matching:

$$\mathcal{L}_{Dist} = Dist(\bigtriangledown_\theta \mathcal{L}_{CE}^{\tilde{D}^g}(\theta), \overline{\bigtriangledown_\theta \mathcal{L}_{CE}^{\tilde{D}^c}}(\theta)), \tag{5}$$

where $\tilde{D}^c$ and $\tilde{D}^g$ denote local and global virtual data, and $\overline{\bigtriangledown_\theta \mathcal{L}_{CE}^{\tilde{D}^c}}$ is the average client gradient. The $Dist(A, B)$ is defined as

$$Dist(A, B) = \sum_{l=1}^{L} \sum_{i=1}^{d_l} (1 - \frac{A_i^l \cdot B_i^l}{||A_i^l|| \, ||B_i^l||}) \tag{6}$$

where $L$ is the number of layers and $A_i^l$ and $B_i^l$ are flattened vectors of gradients corresponding to each output node $i$ from layer $l$ and $d_l$ is the layer output dimension. Then, our proposed federated gradient matching optimize as follows:

$$\min_{D^g} \mathcal{L}_{Dist}(\theta) \quad \text{subject to} \quad \theta = \frac{1}{N}\theta^{c_i{}^*},$$

where $\theta^{c_i{}^*} = \arg\min_\theta \mathcal{L}_i(\tilde{D}^c)$ is the optimal local model weights of client $i$ at a certain round $t$.

Noting that compared with FedAvg (McMahan et al., 2017), there is no additional client information shared for global distillation. We also note the approach seems similar to the gradient inversion attack (Zhu et al., 2019) but we consider averaged gradients w.r.t. local virtual data, and the method potentially [defends] inference attack better (Appendix E.6), which is also implied by (Xiong et al., 2022; Dong et al., 2022). Privacy preservation can be further improved by employing differential privacy (Abadi et al., 2016) in dataset distillation, but this is not the main focus of our work.

## 4 EXPERIMENT

To evaluate FEDLGD, we consider the FL setting in which clients obtain data from different domains while performing the same task. Specifically, we compare with multiple baselines on **benchmark**

**datasets** DIGITS (Sec. 4.2), where each client has data from completely different open-sourced datasets. The experiment is designed to show that FEDLGD can effectively mitigate large domain shifts. Additionally, we evaluate the performance of FEDLGD on another **large benchmark dataset**, CIFAR10C (Hendrycks & Dietterich, 2019), which collects data from different corrupts yielding data distribution shift and contains a large number of clients, so that we can investigate varied client sampling in FL. The experiment aims to show FEDLGD's feasibility on large-scale FL environments. We also validate the performance under **real medical datasets**, RETINA, in Appendix. C.

## 4.1 TRAINING AND EVALUATION SETUP

**Model architecture.** We conduct the ablation study to explore the effect of different deep neural networks' performance under FEDLGD. Specifically, we adapt ResNet18 (He et al., 2016) and ConvNet (Zhao et al., 2021) in our study. To achieve the optimal performance, we apply the same architecture to perform both the local distillation task and the classification task, as this combination is justified to have the best output (Zhao et al., 2021; Zhao & Bilen, 2023). The detailed model architectures are presented in Appendix E.4.

**Comparison methods.** We compare the performance of downstream classification tasks using state-of-the-art (SOTA) FL algorithms, FedAvg (McMahan et al., 2017), FedProx[ (Li et al., 2020a)], FedNova (Wang et al., 2020), Scaffold (Karimireddy et al., 2020), MOON (Li et al., 2021b), and VHL (Tang et al., 2022)[2]. We directly use local virtual data from our initialization stage for FL methods other than ours. We perform classification on client's testing set and report the test accuracies.

**FL training setup.** We use the SGD optimizer with a learning rate of $10^{-2}$ for DIGITS and CIFAR10C. If not specified, our default setting for local model update epochs is 1, total update rounds is 100, the batch size for local training is 32, and the number of virtual data update iterations ($|\tau|$) is 10. The numbers of default virtual data distillation steps for clients and server are set to 100 and 500, respectively. Since we only have a few clients for DIGITS, we will select all the clients for each iteration, while the client selection for CIFAR10C experiments will be specified in Sec. 4.3.

**Proper Initialization for Distillation.** We propose to initialize the virtual data using statistics from local data to take care of both privacy concerns and model performance. Specifically, each client calculates the statistics of its own data for each class, denoted as $\mu_i^c, \sigma_i^c$, and then initializes the distillation images per class, $x \sim \mathcal{N}(\mu_i^c, \sigma_i^c)$, where $c$ and $i$ represent each client and categorical label. For privacy preservation, we use random noise as initialization for global virtual data. The comparison results using different initialization methods proposed in previous works can be found in Appendix D.

## 4.2 DIGITS EXPERIMENT

**Datasets.** We use the following datasets for our benchmark experiments: DIGITS = {MNIST (LeCun et al., 1998), SVHN (Netzer et al., 2011), USPS (Hull, 1994), SynthDigits (Ganin & Lempitsky, 2015), MNIST-M (Ganin & Lempitsky, 2015)}. Each dataset in DIGITS contains handwritten, real street and synthetic digit images of $0, 1, \cdots, 9$. As a result, we have 5 clients in the experiments.

**Comparison with baselines under various conditions.** To validate the effectiveness of FEDLGD, we first compare it with the alternative FL methods varying on two important factors: Image-per-class (IPC) and different deep neural network architectures (arch). We use IPC $\in \{10, 50\}$ and arch $\in$ { ResNet18(R), ConvNet(C)} to examine the performance of SOTA models and FEDLGD using distilled DIGITS. Note that we fix IPC = 10 for global virtual data and vary IPC for local virtual data. Tab. 1 shows the test accuracies of DIGITS experiments. In addition to testing with original test sets, we also show the unweighted averaged test accuracy. One can observe that for each FL algorithm, ConvNet(C) always has the best performance under all IPCs. The observation is consistent with (Zhao & Bilen, 2023) as more complex architectures may cause over-fitting to virtual data. It is also shown that using IPC = 50 always outperforms IPC = 10 as expected since more data are available for training. Overall, FEDLGD outperforms other SOTA methods, where on average accuracy, FEDLGD increases the best test accuracy results among the baseline methods of 2.1% (IPC =10, arch = C), 10.4% (IPC =10, arch = R), 2.2% (IPC = 50, arch = C) and 3.9% (IPC =50, arch = R). VHL (Tang et al., 2022) is the closest strategy to FEDLGD and achieves the best performance among the baseline

---

[2]The detailed information of the methods can be found in Appendix F.

Table 2: Averaged test accuracy for `CIFAR10C` with ConvNet.

| CIFAR10C | | FedAvg | | FedProx | | FedNova | | Scaffold | | MOON | | VHL | | FEDLGD | |
|---|---|---|---|---|---|---|---|---|---|---|---|---|---|---|---|---|
| IPC | | 10 | 50 | 10 | 50 | 10 | 50 | 10 | 50 | 10 | 50 | 10 | 50 | 10 | 50 |
| Client ratio | 0.2 | 27.0 | 44.9 | 27.0 | 44.9 | 26.7 | 34.1 | 27.0 | 44.9 | 20.5 | 31.3 | 21.8 | 45.0 | **32.9** | **46.8** |
| | 0.5 | 29.8 | 51.4 | 29.8 | 51.4 | 29.6 | 45.9 | 30.6 | 51.6 | 23.8 | 43.2 | 29.3 | 51.7 | **39.5** | **52.8** |
| | 1 | 33.0 | 54.9 | 33.0 | 54.9 | 30.0 | 53.2 | 33.8 | 54.5 | 26.4 | 51.6 | 34.4 | 55.2 | **47.6** | **57.4** |

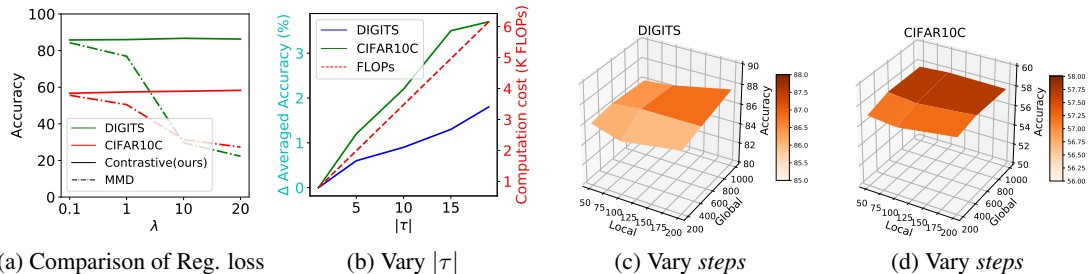

| (a) Comparison of Reg. loss | (b) Vary $|\tau|$ | (c) Vary *steps* | (d) Vary *steps* |
|---|---|---|---|

Figure 3: (a) Comparison between different regularization losses and their weighting in total loss ($\lambda$). One can observe that supervised contrastive loss gives us better and more stable performance with different coefficient choices. (b) The trade-off between $|\tau|$ and computation cost. One can observe that the model performance improves with the increasing $|\tau|$, which is a trade-off between computation cost and model performance. Vary data updating *steps* for (c) `DIGITS` and (d) `CIFAR10C`. One can observe that FEDLGD yields consistent performance, and the accuracy tends to improve with an increasing number of local and global steps.

methods, indicating that the feature alignment solutions are promising for handling heterogeneity in federated virtual learning. However, VHL is still worse than FEDLGD, and the performance may result from the differences in synthesizing global virtual data. VHL (Tang et al., 2022) uses untrained StyleGAN (Karras et al., 2019) to generate global virtual data without further updating. On the contrary, we update our global virtual data during FL training.

### 4.3 `CIFAR10C` EXPERIMENT

**Datasets.** We conduct large-scale FL experiments on `CIFAR10C`[3], where, like previous studies (Li et al., 2021b), we apply Dirichlet distribution with $\alpha = 2$ to generate 3 partitions on each distorted Cifar10-C (Hendrycks & Dietterich, 2019), resulting in 57 *domain and label heterogeneous* non-IID clients. In addition, we apply random client selection with ratio = 0.2, 0.5, and 1

**Comparison with baselines under different client sampling ratios.** The objective of the experiment is to test FEDLGD under popular FL questions: class imbalance, large number of clients, different client sample ratios, and domain and label heterogeneity. One benefit of federated virtual learning is that we can easily handle class imbalance by distilling the same number (IPC) of virtual data. We will vary IPC and fix the model architecture to ConvNet since it is validated that ConvNet yields better performance in virtual training (Zhao et al., 2021; Zhao & Bilen, 2023). One can observe from Tab. 2 that FEDLGD consistently achieves the best performance under different IPC and client sampling ratios. We would like to point out that when IPC=10, the performance boosts are significant, which indicates that FEDLGD is well-suited for FL conditions when there is a large group of clients and each of them has a limited number of data.

### 4.4 ABLATION STUDIES FOR FEDLGD

The success of FEDLGD relies on the novel design of local-global data distillation, where the selection of regularization loss and the number of iterations for data distillation plays a key role. In this section, we study the choice of regularization loss and its weighting ($\lambda$) in the total loss function. Recall that among the total FL training epochs, we perform local-global distillation on the selected $\tau$ *iterations*, and within each selected *iteration*, the server and clients will perform data updating for some pre-defined *steps*. The effect of local-global distillation *iterations* and data updating *steps* will

---

[3]Cifar10-C is a collection of augmented Cifar10 that applies 19 different corruptions.

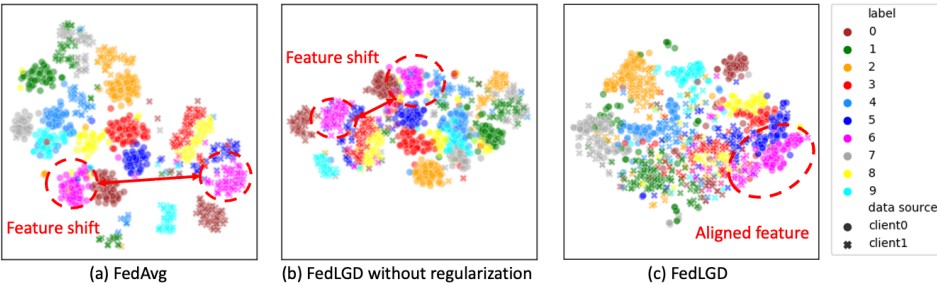

Figure 4: tSNE plots on feature space for FedAvg, FEDLGD without regularization, and FEDLGD. One can observe regularizing training with our global virtual data can rectify feature shifts among different clients.

also be discussed. We use ConvNet and IPC=50 as our default setting. We also perform additional ablation studies such as computation cost and communication overhead in Appendix D.

**Effect of regularization loss.** FEDLGD uses supervised contrastive loss $\mathcal{L}_{\mathrm{Con}}$ as a regularization term to encourage local and global virtual data embedding into a similar feature space. To demonstrate the effectiveness of the regularization term in FEDLGD, we perform ablation studies to replace $\mathcal{L}_{\mathrm{Con}}$ with an alternative distribution similarity measurement, MMD loss, with different $\lambda$'s ranging from 0.1 to 20. Fig. 3a shows the average test accuracy. Using supervised contrastive loss gives us better and more stable performance with different coefficient choices. To explain the effect of our proposed regularization loss on feature representations, we embed the latent features before fully-connected layers to a 2D space using tSNE (Van der Maaten & Hinton, 2008) shown in Fig. 4. For the model trained with FedAvg (Fig. 4a), features from two clients ($\times$ and $\circ$) are closer to their own distribution regardless of the labels (colors). In Fig. 4b, we perform virtual FL training but without the regularization term (Eq. 4). Fig. 4c shows FEDLGD, and one can observe that data from different clients with the same label are grouped together. This indicates that our regularization with global virtual data is useful for learning homogeneous feature representations.

**Analysis of distillation *iterations* ($|\tau|$).** Fig. 3b shows the improved averaged test accuracy if we increase the number of distillation iterations with FEDLGD. The base accuracy for DIGITS and CIFAR10C are 85.8 and 55.2, respectively. We fix local and global update *steps* to 100 and 500, and the selected iterations ($\tau$) are defined as arithmetic sequences with $d = 5$ (i.e., $\tau = \{0, 5, ...\}$). One can observe that the model performance improves with the increasing $|\tau|$. This is because we obtain better virtual data with more local-global distillation iterations, which is a trade-off between computation cost and model performance. We select $|\tau| = 10$ for efficiency trade-off.

**Robustness on virtual data update *steps*.** In Fig. 3c and Fig. 3d, we fix $|\tau| = 10$, and vary (local, global) data updating steps. One can observe that FEDLGD yields stable performance, and the accuracy slightly improves with an increasing number of local and global steps. Nevertheless, the results are all the best when comparing with the baselines. It is also worth noting that there is still trade-off between *steps* and computation cost (See Appendix).

## 5 CONCLUSION

In this paper, we introduce a new approach for FL, called FEDLGD. It utilizes virtual data on both client and server sides to train FL models. We are the first to reveal that FL on local virtual data can increase heterogeneity. Furthermore, we propose iterative distribution matching and federated gradient matching to iteratively update local and global virtual data, and apply global virtual regularization to effectively harmonize domain shift. Our experiments on benchmark and real medical datasets show that FEDLGD outperforms current state-of-the-art methods in heterogeneous settings. Furthermore, FEDLGD can be combined with other heterogenous FL methods such as FedProx[ (Li et al., 2020a)] and Scaffold (Karimireddy et al., 2020) to further improve its performance. The potential limitation lies in the additional communication and computation cost in data distillation, but we show that the trade-off is acceptable and can be mitigated by decreasing distillation *iterations* and *steps*. Our future direction will be investigating privacy-preserving data generation. We believe that this work sheds light on how to effectively mitigate data heterogeneity from a dataset distillation perspective and will inspire future work to enhance FL performance, privacy, and efficiency.

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

**Road Map of Appendix** Our appendix is organized into five sections. The notation table is in Appendix A, which contains the mathematical notation and Algorithm 1, which outlines the pipeline of FEDLGD. Appendix C shows the results for RETINA, a real-world medical dataset. Appendix D provides a list of ablation studies to analyze FEDLGD, including computation cost, communication overhead, convergence rate, and hyper-parameter choices. Appendix E lists the details of our experiments: E.1 visualizes the original sample images used in our experiments; E.2 visualizes the local and global distilled images; E.3 shows the pixel histogram for the DIGITS and RETINA datasets for visualizing the heterogeneity of them; E.4 shows the model architectures that we used in the experiments; E.5 contains the hyper-parameters that we used to conduct all experiments; E.6 provides experiments and analysis for the privacy of FEDLGD through membership inference attack. Finally, Appendix F provides a detailed literature review and implementation of the state-of-the-art heterogeneous FL strategies. Our code and model checkpoints are available in this anonymous link: https://drive.google.com/drive/folders/1Hpy8kgPtxC_NMqK6eALwukFZJB7yf8Vl?usp=sharing[4].

## A  NOTATION TABLE

Table 3: Important notations used in the paper.

| Notations | Description |
|---|---|
| $d$ | input dimension |
| $d'$ | feature dimension |
| $f^\theta$ | global model |
| $\theta$ | model parameters |
| $\psi$ | feature extractor |
| $h$ | projection head |
| $D^g, D^c$ | original global and local data |
| $\tilde{D}^g, \tilde{D}^c$ | global and local synthetic data |
| $\tilde{f}^g, \tilde{f}^c$ | features of global and local synthetic data |
| $\mathcal{L}_{\text{total}}$ | total loss function for virtual federated training |
| $\mathcal{L}_{\text{CE}}$ | cross-entropy loss |
| $\mathcal{L}_{\text{Dist}}$ | Distance loss for gradient matching |
| $\mathcal{L}_{\text{MMD}}$ | MMD loss for distribution matching |
| $\mathcal{L}_{\text{Con}}$ | Contrastive loss for local training regularization |
| $\lambda$ | coefficient for local training regularization term |
| $T$ | total training iterations |
| $T_{\text{D}}^{\text{c}}$ | local data updating iterations for each call |
| $T_{\text{D}}^{\text{g}}$ | global data updating iterations for each call |
| $\tau$ | local global distillation iterations |

---

[4]The link was created by a new and anonymous account without leaking any identifiable information.

---

**Algorithm 1** Federated Virtual Learning with Local-global Distillation

---

**Require:** $f^\theta$: Model, $\psi^\theta$: Feature extractor, $\theta$: Model parameters, $\tilde{D}$: Virtual data, $D$: Original data, $\mathcal{L}$: Losses, $G$: Gradients.

**Distillation Functions:**
$\tilde{D}^{\mathrm{c}} \leftarrow \mathrm{DistributionMatch}(D^{\mathrm{c}}, f^\theta)$
$\tilde{D}^{\mathrm{c}}_{\mathrm{t}} \leftarrow \mathrm{IterativeDistributionMatch}(\tilde{D}^{\mathrm{c}}_{\mathrm{t-1}}, f^\theta_{\mathrm{t}})$
$\tilde{D}^{\mathrm{g}}_{\mathrm{t+1}} \leftarrow \mathrm{FederatedGradientMatch}(\tilde{D}^{\mathrm{g}}_{\mathrm{t}}, G^{\mathrm{g}}_{\mathrm{t}})$

**Initialization:**
$\tilde{D}^{\mathrm{c}}_{0} \leftarrow \mathrm{DistributionMatch}(D^{\mathrm{c}}_{\mathrm{rand}}, f^\theta_{\mathrm{rand}})$          ▷ Distilled local data for virtual FL training

**FEDLGD Pipeline:**
**for** $t = 1, \ldots, T$ **do**
    **Clients:**
    **for** each selected Client **do**
        **if** $t \in \tau$ **then**                                       ▷ Local-global distillation
            $\tilde{D}^{\mathrm{c}}_{\mathrm{t}} \leftarrow \mathrm{IterativeDistributionMatch}(\tilde{D}^{\mathrm{c}}_{\mathrm{t-1}}, f^\theta_{\mathrm{t}})$
            $G^{\mathrm{c}}_{\mathrm{t}} \leftarrow \nabla_\theta \mathcal{L}_{\mathrm{CE}}(\tilde{D}^{\mathrm{c}}_{\mathrm{t}}, f^\theta_{\mathrm{t}})$
        **else**
            $\tilde{D}^{\mathrm{c}}_{\mathrm{t}} \leftarrow \tilde{D}^{\mathrm{c}}_{\mathrm{t-1}}$
            $G^{\mathrm{c}}_{\mathrm{t}} \leftarrow \nabla_\theta \left( \mathcal{L}_{\mathrm{CE}}(\tilde{D}^{\mathrm{c}}_{\mathrm{t}}, f^\theta_{\mathrm{t}}) + \lambda \mathcal{L}_{\mathrm{CON}}(\psi^\theta_{\mathrm{t}}(\tilde{D}^{\mathrm{g}}_{\mathrm{t}}), \psi^\theta_{\mathrm{t}}(\tilde{D}^{\mathrm{c}}_{\mathrm{t}})) \right)$
        **end if**
        Uploads $G^{\mathrm{c}}_t$ to Server
    **end for**
    **Server:**
    $G^{\mathrm{g}}_{\mathrm{t}} \leftarrow \mathrm{Aggregate}(G^{1}_{\mathrm{t}}, ..., G^{\mathrm{c}}_{\mathrm{t}})$
    **if** $t \in \tau$ **then**                                   ▷ Local-global distillation
        $\tilde{D}^{\mathrm{g}}_{\mathrm{t+1}} \leftarrow \mathrm{FederatedGradientMatch}(\tilde{D}^{\mathrm{g}}_{\mathrm{t}}, G^{\mathrm{g}}_{\mathrm{t}})$
        Send $\tilde{D}^{\mathrm{g}}_{\mathrm{t+1}}$ to Clients
    **end if**
    $f^\theta_{\mathrm{t+1}} \leftarrow \mathrm{ModelUpdate}(G^{\mathrm{g}}_{\mathrm{t}}, f^\theta_{\mathrm{t}})$
    Send $f^\theta_{\mathrm{t+1}}$ to Clients
**end for**

---

# B    THEORETICAL ANALYSIS

In this section, we show theoretical insights on FEDLGD.

Denote the distribution of global virtual data as $\mathcal{P}_g$ and the distribution of client local virtual data as $\mathcal{P}_c$. In providing theoretical justification for the efficacy of FEDLGD, we can adopt a similar analysis approach as demonstrated in Theorem 3.2 of VHL (Tang et al., 2022), where the relationship between generalization performance and domain misalignment for classification tasks is studied by considering *maximizing* the statistic margin (SM) (Koltchinskii & Panchenko, 2002).

To assess the generalization performance of $f$ with respect to the distribution $\mathcal{P}(x, y)$, we define the SM of FEDLGD as follows:

$$\mathbb{E}_{f=\text{FEDLGD}(\mathcal{P}_g(x,y))} SM_m(f, \mathcal{P}(x,y)), \tag{7}$$

where $m$ is a distance metric, and $f = \text{FEDLGD}(\mathcal{P}_g(x, y))$ means that model $f$ is optimized using FEDLGD with minimizing Eq. 3. Similar to Theorem A.2 of (Tang et al., 2022), we have the lower bound

**Lemma B.1** (Lower bound of FEDLGD's statistic margin). *Let $f = \phi \circ \rho$ be a neural network decompose of a feature extractor $\phi$ and a classifier $\rho$. The lower bound of* FEDLGD*'s SM is*

$$\mathbb{E}_{\rho \leftarrow \mathcal{P}_g} SM_m(\rho, \mathcal{P}) \geq \mathbb{E}_{\rho \leftarrow \mathcal{P}_g} SM_m(\rho, \tilde{D}) - \left| \mathbb{E}_{\rho \leftarrow \mathcal{P}_g} \left[ SM_m(\rho, \mathcal{P}_g) - SM_m(\rho, \tilde{D}) \right] \right|$$
$$- \mathbb{E}_y d\left( \mathcal{P}_c(\phi \mid y), \mathcal{P}_g(\phi \mid y) \right). \tag{8}$$

*Proof.* Following proof in Theorem A.2 of (Tang et al., 2022), the statistical margin is decomposed as

$$\mathbb{E}_{\rho \leftarrow \mathcal{P}_g} SM_m(\rho, \mathcal{P}) \geq \mathbb{E}_{\rho \leftarrow \mathcal{P}_g} SM_m(\rho, \tilde{D})$$
$$- \left| \mathbb{E}_{\rho \leftarrow \mathcal{P}_g} \left[ SM_m(\rho, \mathcal{P}_g) - SM_m(\rho, \tilde{D}) \right] \right|$$
$$- \left| \mathbb{E}_{\rho \leftarrow \mathcal{P}_g} \left[ SM_m(\rho, \mathcal{P}) - SM_m(\rho, \mathcal{P}_g) \right] \right|$$
$$\geq \mathbb{E}_{\rho \leftarrow \mathcal{P}_g} SM_m(\rho, \tilde{D}) - \left| \mathbb{E}_{\rho \leftarrow \mathcal{P}_g} \left[ SM_m(\rho, \mathcal{P}_g) - SM_m(\rho, \tilde{D}) \right] \right|$$
$$- \mathbb{E}_y d\left( \mathcal{P}(\phi \mid y), \mathcal{P}_g(\phi \mid y) \right)$$

$\square$

Another component in our analysis is building the connection between our used gradient matching strategy and the distribution match term in the bound.

**Lemma B.2** (Proposition 2 of (Yu et al., 2023)). *First-order distribution matching objective is approximately equal to gradient matching of each class for kernel ridge regression models following a random feature extractor.*

**Theorem B.3.** *Due to the complexity of data distillation steps, without loss of generality, we consider kernel ridge regression models with a random feature extractor. Minimizing total loss of* FEDLGD *(Eq. 2) for harmonizing local heterogeneity with global anchors elicits a model with bounded statistic margin (i.e., the upper bound of the SM bound in Theorem B.1).*

*Proof.* The first and second term can be bounded by maximizing SM of local virtual training data and global virtual data. The large SM of global virtual data distribution $\mathcal{P}_g(x, y)$ is encouraged by minimizing cross-entropy $L_{CE}(\tilde{D}^g, y)$ in our objective function Eq. 3.

The third term represents the discrepancy of distributions of virtual and real data. We denote this term as $\mathcal{D}_{\phi|y}^{\mathcal{P}_c}(\mathcal{P}_g) = \mathbb{E}_y d\left( \mathcal{P}_c(\phi \mid y), \mathcal{P}_g(\phi \mid y) \right)$ and aim to show that $\mathcal{D}_{\phi|y}^{\mathcal{P}_c}(\mathcal{P}_g)$ can achieve small upper bound under proper assumptions.

Based on Lemma B.2, the first-order distribution matching objective $\mathcal{D}_{\phi|y}^{\mathcal{P}_c}(\mathcal{P}_g)$ is approximately equal to gradient matching of each class, as shown in objective $\mathcal{L}_{Dist}$ (Eq. 5). Namely, minimizing gradient matching objective $\mathcal{L}_{Dist}$ in FEDLGD implies minimizing $\mathcal{D}_{\phi|y}^{\mathcal{P}_c}(\mathcal{P}_g)$ in the setting. Hence, using gradient matching generated global virtual data elicits the model's SM a tight lower bound.

$\square$

*Remark* B.4. The key distinction between FEDLGD and VHL primarily lies in the final term, which is exactly a distribution matching objective. It is important to note that in VHL, the global virtual data is generated from an un-pretrained StyleGAN, originating from various Gaussian distributions, which we denote as $\mathcal{P}_g$. The VHL paper only provided a lower bound for $\mathcal{D}_{\phi|y}^{\mathcal{P}_c}(\mathcal{P}_g)$ but did not show how it is upper bounded. However, for the purpose of maximizing SM to achieve strong generalization, we want to show SM has a tight lower bound. Therefore, upper bounded the last term is desired. In contrast, our approach employs the *gradient matching* strategy to synthesize the global virtual data. To prove our performance improvement, we can show that FEDLGD could achieve a tight lower bound for SM.

# C Experiment Results on Real-world Dataset

Table 4: Test accuracy for RETINA experiments under different model architectures and IPC=10. R and C stand for ResNet18 and ConvNet, respectively. We have 4 clients: Drishti(D), Acrima(A), Rim(Ri), and Refuge(Re), respectively. We also show the average test accuracy (Avg). The best results on ConvNet are marked in red and in **bold** for ResNet18. The same accuracy for different methods is due to the limited number of testing samples.

| RETINA | | D | A | Ri | Re | Avg |
|---|---|---|---|---|---|---|
| FedAvg | R | 31.6 | 71.0 | 52.0 | **78.5** | 58.3 |
| | C | 69.4 | 84.0 | **88.0** | 86.5 | 82.0 |
| FedProx | R | 31.6 | 70.0 | 52.0 | **78.5** | 58.0 |
| | C | 68.4 | 84.0 | **88.0** | 86.5 | 81.7 |
| FedNova | R | 31.6 | 71.0 | 52.0 | **78.5** | 58.3 |
| | C | 68.4 | 84.0 | **88.0** | 86.5 | 81.7 |
| Scaffold | R | 31.6 | 73.0 | 49.0 | **78.5** | 58.0 |
| | C | 68.4 | 84.0 | **88.0** | 86.5 | 81.7 |
| MOON | R | 42.1 | 71.0 | 57.0 | 70.0 | 60.0 |
| | C | 57.9 | 72.0 | 76.0 | 85.0 | 72.7 |
| VHL | R | 47.4 | 62.0 | 50.0 | 76.5 | 59.0 |
| | C | 68.4 | 78.0 | 81.0 | 87.0 | 78.6 |
| FedLGD | R | **57.9** | **75.0** | **59.0** | 77.0 | **67.2** |
| | C | **78.9** | **86.0** | **88.0** | **87.5** | **85.1** |

**Dataset.** For medical dataset, we use the retina image datasets, RETINA = {Drishti (Sivaswamy et al., 2014), Acrima(Diaz-Pinto et al., 2019), Rim (Batista et al., 2020), Refuge (Orlando et al., 2020)}, where each dataset contains retina images from different stations with image size $96 \times 96$, thus forming four clients in FL. We perform binary classification to identify *Glaucomatous* and *Normal*. Example images and distributions can be found in Appendix E.3. Each client has a held-out testing set. In the following experiments, we will use the distilled local virtual training sets for training and test the models on the original testing sets. The sample population statistics for both experiments are available in Table 12 and Table 14 in Appendix E.5.

**Comparison with baselines.** The results for RETINA experiments are shown in Table 4, where D, A, Ri, Re represent Drishti, Acrima, Rim, and Refuge datasets. We only set IPC=10 for this experiment as clients in RETINA contain much fewer data points. The learning rate is set to 0.001. The same as in the previous experiment, we vary arch $\in$ { ConvNet, ResNet18}. Similarly, ConvNet shows the best performance among architectures, and FedLGD has the best performance compared to the other methods w.r.t the unweighted averaged accuracy (Avg) among clients. To be precise, FedLGD increases unweighted averaged test accuracy for 3.1%(versus the best baseline) on ConvNet and 7.2%(versus the best baseline) on ResNet18, respectively. The same accuracy for different methods is due to the limited number of testing samples. We conjecture the reason why VHL (Tang et al., 2022) has lower performance improvement in RETINA experiments is that this dataset is in higher dimensional and clinical diagnosis evidence on fine-grained details, *e.g.*, cup-to-disc ratio and disc rim integrity (Schuster et al., 2020). Therefore, it is difficult for untrained StyleGAN (Karras et al., 2019) to serve as anchor for this kind of larger images.

# D  ADDITIONAL RESULTS AND ABLATION STUDIES FOR FEDLGD

## D.1  DIFFERENT RANDOM SEEDS

To show the consistent performance of FEDLGD, we repeat the experiments for DIGITS, CIFAR10C, and RETINA with three random seeds, and report the validation loss and accuracy curves in Figure 5 and 6 (The standard deviations of the curves are plotted as shadows.). We use ConvNet for all the experiments. IPC is set to 50 for CIFAR10C and DIGITS; 10 for RETINA. We use the default hyperparameters for each dataset, and only report FedAvg, FedProx, Scaffold, VHL, which achieves the best performance among baseline as indicated in Table 1, 2, and 4 for clear visualization. One can observe that FEDLGD has faster convergence rate and results in optimal performances compared to other baseline methods.

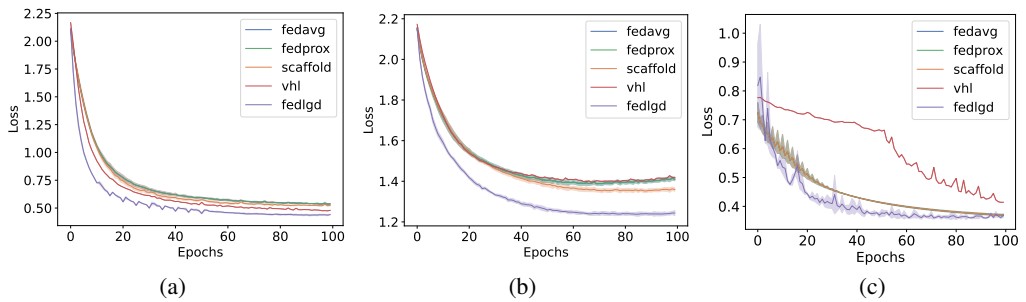

|     |     |     |
| (a) | (b) | (c) |

Figure 5: Averaged testing loss for (a) DIGITS with IPC = 50, (b) CIFAR10C with IPC = 50, and (c) RETINA with IPC = 10 experiments.

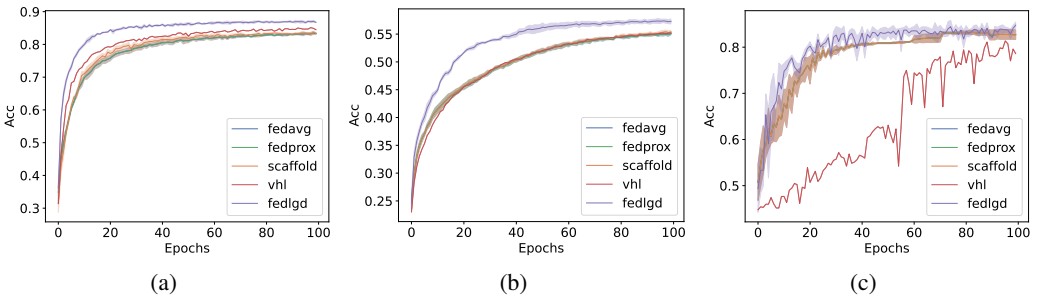

|     |     |     |
| (a) | (b) | (c) |

Figure 6: Averaged testing accuracy for (a) DIGITS with IPC = 50, (b) CIFAR10C with IPC = 50, and (c) RETINA with IPC = 10 experiments.

## D.2  DIFFERENT HETEROGENEITY LEVELS OF LABEL SHIFT

In the experiment presented in Sec 4.3, we study FEDLGD under both label and domain shifts, where labels are sampled from Dirichlet distribution. To ensure dataset distillation performance, we ensure that each class at least has 100 samples per client, thus setting the coefficient of Dirichlet distribution $\alpha = 2$ to simulate the worst case of label heterogeneity that meets the quality dataset distillation requirement. Here, we show the performance with a less heterogeneity level ($\alpha = 5$) while keeping the other settings the same as those in Sec.4.3. The results are shown in Table 5. As we expect, the performance drop when the heterogeneity level increases ($\alpha$ decreases). One can observe that when heterogeneity increases, FEDLGD's performance drop less except for VHL. We conjecture that VHL yields similar test accuracy for $\alpha = 2$ and $\alpha = 5$ is that it uses fixed global virtual data so that the effectiveness of regularization loss does not improve much even if the heterogeneity level is decreased. Nevertheless, FEDLGD consistently outperforms all the baseline methods.

Table 5: Comparison of different $\alpha$ for Drichilet distribution on `CIFAR10C`.

| $\alpha$ | FedAvg (McMahan et al., 2017) | FedProx (Li et al., 2020b) | FedNova (Wang et al., 2020) | Scaffold (Karimireddy et al., 2020) | MOON (Li et al., 2021b) | VHL (Tang et al., 2022) | FedLGD |
|---|---|---|---|---|---|---|---|
| 2 | 54.9 | 54.9 | 53.2 | 54.5 | 51.6 | 55.2 | **57.4** |
| 5 | 55.4 | 55.4 | 55.4 | 55.6 | 51.1 | 55.4 | **58.1** |

Table 6: Computation cost for each epoch. Nc and Ns stand for the number of updating iteration for local and global virtual data, and we defaultly set as 100 and 500, respectively. Note that we only set $|\tau| = 10$ iterations, which is a relatively small number compare to total epochs(100).

| Dataset | Vanilla FedAvg | FedLGD(iters $\in \tau$) | FedLGD(iters $\notin \tau$) | FedLGD(server) |
|---|---|---|---|---|
| `DIGITS` | 238K | 2.7K + 3.4K × Nc | 4.8K | 2.9K × Ns |
| `CIFAR10C` | 53M | 2.7K + 3.4K × Nc | 4.8K | 2.9K × Ns |
| `RETINA` | 1.76M | 0.7K + 0.9K × Nc | 1K | 0.9K × Ns |

## D.3 COMPUTATION COST

Computation cost for `DIGITS` experiment on each epoch can be found in [Table 6]. Nc and Ns stand for the number of updating iterations for local and global virtual data, and as default, we it set as 100 and 500, respectively. The computation costs for FedLGD in `DIGITS` and `CIFAR10C` are identical since we used virtual data with fixed size and number for training. Plugging in the number, clients only need to operate 3.9M FLOPs for total 100 training epochs with $\tau = 10$ (our default setting), which is significantly smaller than vanilla FedAvg using original data (23.8M and [53M] for `DIGITS` and `CIFAR10C`, respectively.).

Table 7: Communication overhead for each epoch. Note that the IPC for our global virtual data is 10, and the clients only need to *download* it for $|\tau| = 10$ times.

| Image size | ConvNet | ResNet18 | Global virtual data |
|---|---|---|---|
| $28 \times 28$ | 311K | 11M | 23K × IPC |
| $96 \times 96$ | 336K | 13M | 55K × IPC |

## D.4 COMMUNICATION OVERHEAD

The communication overhead for each epoch in `DIGITS` and `CIFAR10C` experiments are identical since we use same architectures and size of global virtual data (Table. 7 $28 \times 28$). The analysis of `RETINA` is shown in row $96 \times 96$. Note that the IPC for our global virtual data is 10, and the clients only need to *download* it for $|\tau|$ times. Although FedLGD requires clients to download additional data which is almost double the original Bytes (311K + 230K), we would like to point out that this only happens $|\tau| = 10$ times, which is a relatively small number compared to total FL training iterations.

## D.5 ANALYSIS OF BATCH SIZE

Batch size is another factor for training the FL model and our distilled data. We vary the batch size $\in \{8, 16, 32, 64\}$ to train models for `CIFAR10C` with the fixed default learning rate. We show the effect of batch size in Table 8 reported on average testing accuracy. One can observe that the performance is slightly better with moderately smaller batch size which might due to two reasons: 1) more frequent model update locally; and 2) larger model update provides larger gradients, and FedLGD can benefit from the large gradients to distill higher quality virtual data. Overall, the results are generally stable with different batch size choices.

## D.6 ANALYSIS OF LOCAL EPOCH

Aggregating at different frequencies is known as an important factor that affects FL behavior. Here, we vary the local epoch $\in \{1, 2, 5\}$ to train all baseline models on `CIFAR10C`. Figure 7 shows the result of test accuracy under different epochs. One can observe that as the local epoch increases, the

Table 8: Varying batch size in FEDLGD on `CIFAR10C`. We report the unweighted accuracy. One can observe that the performance increases when the batch size decreases.

| Batch Size | 8 | 16 | 32 | 64 |
|---|---|---|---|---|
| CIFAR10C | 59.5 | 58.3 | 57.4 | 56.0 |

performance of FEDLGD would drop a little bit. This is because doing gradient matching requires the model to be trained to an intermediate level, and if local epochs increase, the loss of `DIGITS` models will drop significantly. However, FEDLGD still consistently outperforms the baseline methods. As our future work, we will investigate the tuning of the learning rate in the early training stage to alleviate the effect.

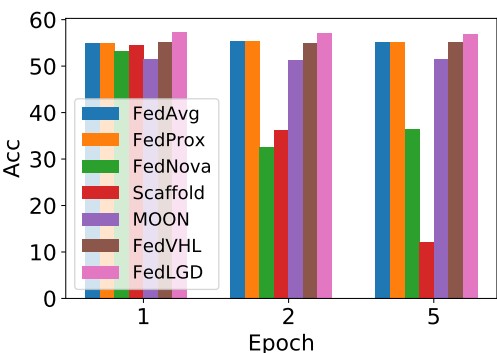

Figure 7: Comparison of model performances under different local epochs with `CIFAR10C`.

### D.7 DIFFERENT INITIALIZATION FOR VIRTUAL IMAGES

To validate our proposed initialization for virtual images has the best trade-off between privacy and efficacy, we compare our test accuracy with the models trained with synthetic images initialized by random noise and real images in Table 9. To show the effect of initialization under large domain shift, we run experiments on `DIGITS` dataset. One can observe that our method which utilizes the statistics $(\mu_i, \sigma_i)$ of local clients as initialization outperforms random noise initialization. Although our performance is slightly worse than the initialization that uses real images from clients, we do not ask the clients to share real images to the server which is more privacy-preserving.

Table 9: Comparison of different initialization for synthetic images `DIGITS`

| DIGITS | MNIST | SVHN | USPS | SynthDigits | MNIST-M | Average |
|---|---|---|---|---|---|---|
| Noise ($\mathcal{N}(0,1)$) | 96.3 | 75.9 | 93.3 | 72.0 | 83.7 | 84.2 |
| Ours ($\mathcal{N}(\mu_i, \sigma_i)$) | 97.1 | 77.3 | 94.6 | 78.5 | 86.1 | 86.7 |
| Real images | 97.7 | 78.8 | 94.2 | 82.4 | 89.5 | 88.5 |

# E EXPERIMENTAL DETAILS

## E.1 VISUALIZATION OF THE ORIGINAL IMAGES

### E.1.1 DIGITS DATASET

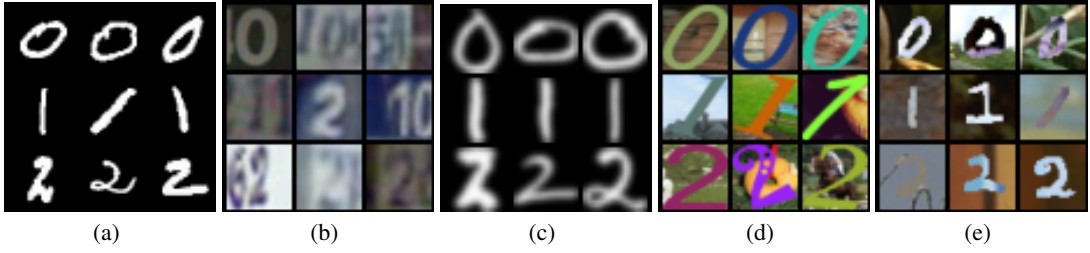

(a)    (b)    (c)    (d)    (e)

Figure 8: Visualization of the original digits dataset. (a) visualized the MNIST client; (b) visualized the SVHN client; (c) visualized the USPS client; (d) visualized the SynthDigits client; (e) visualized the MNIST-M client.

### E.1.2 RETINA DATASET

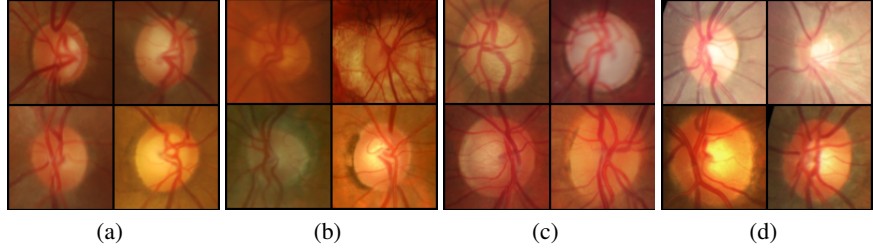

(a)    (b)    (c)    (d)

Figure 9: Visualization of the original retina dataset. (a) visualized the Drishti client; (b) visualized the Acrima client; (c) visualized the Rim client; (d) visualized the Refuge client.

### E.1.3 CIFAR10C DATASET

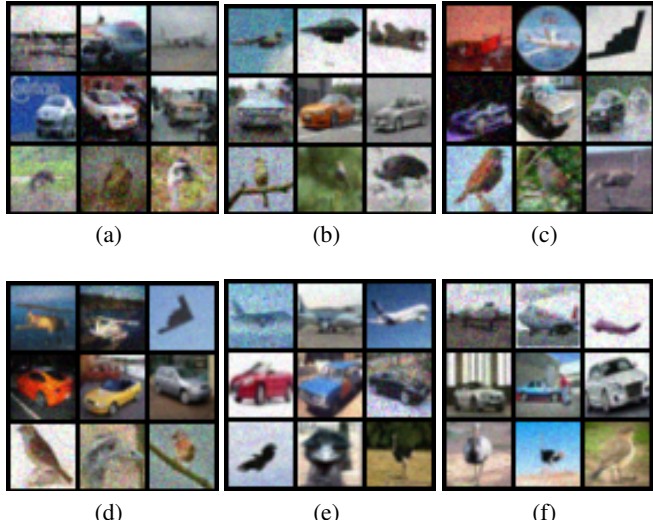

Figure 10: Visualization of the original CIFAR10C. Sampled images from the first six clients.

## E.2   VISUALIZATION OF OUR DISTILLED GLOBAL AND LOCAL IMAGES

### E.2.1   DIGITS DATASET

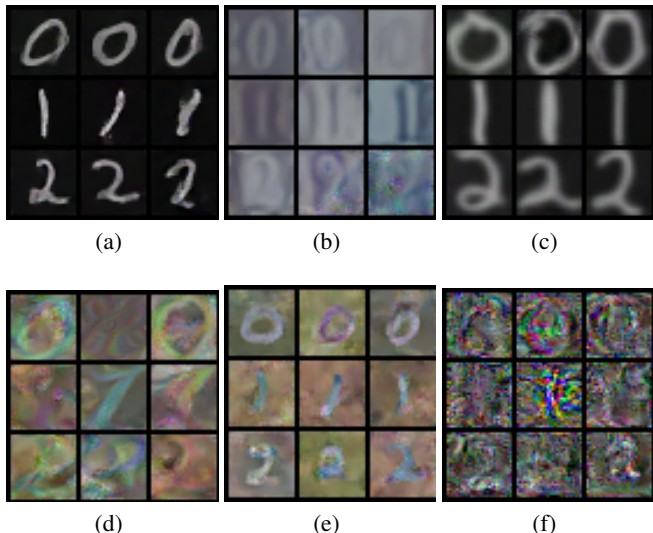

Figure 11: Visualization of the global and local distilled images from the digits dataset. (a) visualized the MNIST client; (b) visualized the SVHN client; (c) visualized the USPS client; (d) visualized the SynthDigits client; (e) visualized the MNIST-M client; (f) visualized the server distilled data.

### E.2.2   RETINA DATASET

### E.2.3   CIFAR10C DATASET

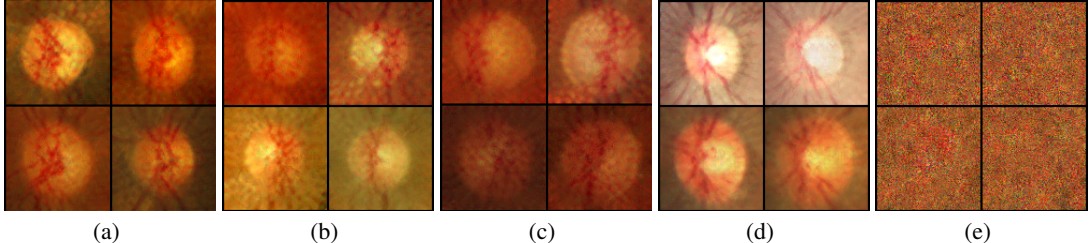

Figure 12: Visualization of the global and local distilled images from retina dataset. (a) visualized the Drishti client; (b) visualized the Acrima client; (c) visualized the Rim client; (d) visualized the Refuge client; (e) visualized the server distilled data.

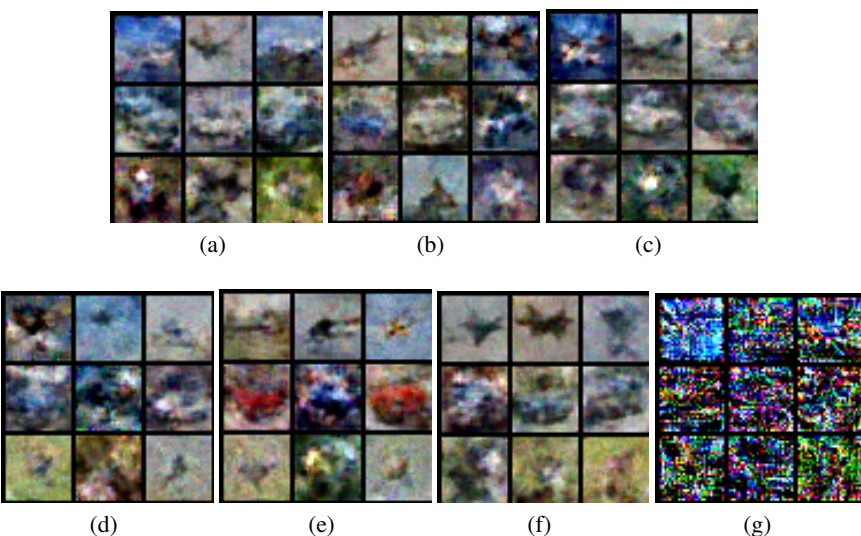

Figure 13: (a)-(f) visualizes the distailled images for the first six clients of CIFAR10C. (g) visualizes the global distilled images.

### E.3 VISUALIZATION OF THE HETEROGENEITY OF THE DATASETS

### E.3.1 DIGITS DATASET

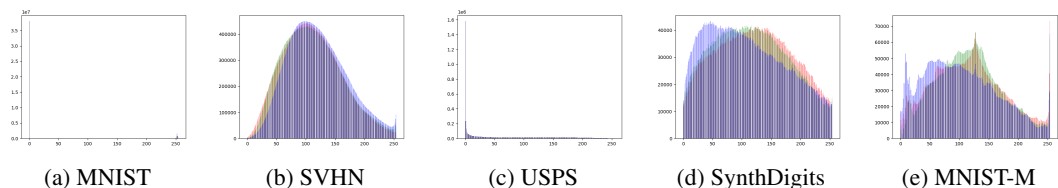

   (a) MNIST       (b) SVHN       (c) USPS    (d) SynthDigits   (e) MNIST-M

Figure 14: Histogram for the frequency of each RGB value in original DIGITS. The red bar represents the count for R; the green bar represents the frequency of each pixel for G; the blue bar represents the frequency of each pixel for B. One can observe the distributions are very different. Note that figure (a) and figure (c) are both greyscale images with most pixels lying in 0 and 255.

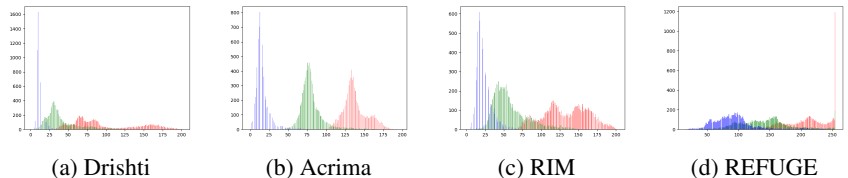

(a) Drishti      (b) Acrima      (c) RIM      (d) REFUGE

Figure 15: Histogram for the frequency of each RGB value in original `RETINA`. The red bar represents the count for R; the green bar represents the frequency of each pixel for G; the blue bar represents the frequency of each pixel for B.

### E.3.2 RETINA DATASET

### E.3.3 CIFAR10C DATASET

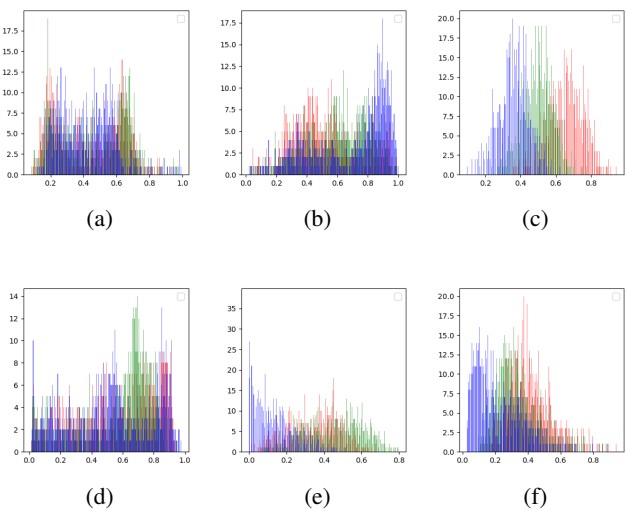

(a)      (b)      (c)

(d)      (e)      (f)

Figure 16: Histogram for the frequency of each RGB value in the first six clients of original `CIFAR10C`. The red bar represents the count for R; the green bar represents the frequency of each pixel for G; the blue bar represents the frequency of each pixel for B.

### E.4 MODEL ARCHITECTURE

For our benchmark experiments, we use ConvNet to both distill the images and train the classifier.

Table 10: ResNet 18 architecture. For the convolutional layer (Conv2D), we list parameters with a sequence of input and output dimensions, kernel size, stride, and padding. For the max pooling layer (MaxPool2D), we list kernel and stride. For a fully connected layer (FC), we list input and output dimensions. For the BatchNormalization layer (BN), we list the channel dimension.

| Layer | Details |
|---|---|
| 1 | Conv2D(3, 64, 7, 2, 3), BN(64), ReLU |
| 2 | Conv2D(64, 64, 3, 1, 1), BN(64), ReLU |
| 3 | Conv2D(64, 64, 3, 1, 1), BN(64) |
| 4 | Conv2D(64, 64, 3, 1, 1), BN(64), ReLU |
| 5 | Conv2D(64, 64, 3, 1, 1), BN(64) |
| 6 | Conv2D(64, 128, 3, 2, 1), BN(128), ReLU |
| 7 | Conv2D(128, 128, 3, 1, 1), BN(64) |
| 8 | Conv2D(64, 128, 1, 2, 0), BN(128) |
| 9 | Conv2D(128, 128, 3, 1, 1), BN(128), ReLU |
| 10 | Conv2D(128, 128, 3, 1, 1), BN(64) |
| 11 | Conv2D(128, 256, 3, 2, 1), BN(128), ReLU |
| 12 | Conv2D(256, 256, 3, 1, 1), BN(64) |
| 13 | Conv2D(128, 256, 1, 2, 0), BN(128) |
| 14 | Conv2D(256, 256, 3, 1, 1), BN(128), ReLU |
| 15 | Conv2D(256, 256, 3, 1, 1), BN(64) |
| 16 | Conv2D(256, 512, 3, 2, 1), BN(512), ReLU |
| 17 | Conv2D(512, 512, 3, 1, 1), BN(512) |
| 18 | Conv2D(256, 512, 1, 2, 0), BN(512) |
| 19 | Conv2D(512, 512, 3, 1, 1), BN(512), ReLU |
| 20 | Conv2D(512, 512, 3, 1, 1), BN(512) |
| 21 | AvgPool2D |
| 22 | FC(512, num_class) |

Table 11: ConvNet architecture. For the convolutional layer (Conv2D), we list parameters with a sequence of input and output dimensions, kernel size, stride, and padding. For the max pooling layer (MaxPool2D), we list kernel and stride. For a fully connected layer (FC), we list the input and output dimensions. For the GroupNormalization layer (GN), we list the channel dimension.

| Layer | Details |
|---|---|
| 1 | Conv2D(3, 128, 3, 1, 1), GN(128), ReLU, AvgPool2d(2,2,0) |
| 2 | Conv2D(128, 118, 3, 1, 1), GN(128), ReLU, AvgPool2d(2,2,0) |
| 3 | Conv2D(128, 128, 3, 1, 1), GN(128), ReLU, AvgPool2d(2,2,0) |
| 4 | FC(1152, num_class) |

## E.5 TRAINING DETAILS

We provide detailed settings for experiments conducted in Table 12 for `DIGITS`, Table 13 for `CIFAR10C`, and Table 14 for `RETINA`. The experiments are run on NVIDIA GeForce RTX 3090 Graphics cards with PyTorch.

Table 12: `DIGITS` settings for all federated learning, including the number of training and testing examples, and local update epochs. Image per class is the number of distilled images used for distribution matching only in FEDLGD. The image size is set to $28 \times 28$.

| DataSets | MNIST | SVHN | USPS | SynthDigits | MNIST-M |
|---|---|---|---|---|---|
| Number of clients | 1 | 1 | 1 | 1 | 1 |
| Number of Training Samples | 60000 | 73257 | 7291 | 10000 | 10331 |
| Number of Testing Samples | 10000 | 26032 | 2007 | 2000 | 209 |
| Image per Class | 10,**50** | 10,**50** | 10,**50** | 10,**50** | 10,**50** |
| Local Update Epochs | **1**,2,5 | **1**,2,5 | **1**,2,5 | **1**,2,5 | **1**,2,5 |
| Local Distillation Update Epochs | 50, **100**, 200 | 50, **100**, 200 | 50, **100**, 200 | 50, **100**, 200 | 50, **100**, 200 |
| global Distillation Update Epochs | 200, **500**, 1000 | 200, **500**, 1000 | 200, **500**, 1000 | 200, **500**, 1000 | 200, **500**, 1000 |
| $\lambda$ | 10 | 10 | 10 | 10 | 10 |

Table 13: `CIFAR10C` settings for all federated learning, including the client ratio for training and testing examples, and local update epochs. Image per class is the number of distilled images used for distribution matching only in FEDLGD. The image size is set to $28 \times 28$.

| $\alpha$ | 2 | 5 |
|---|---|---|
| Number of clients | 57 | 57 |
| Averaged Number of Training Samples | 21790 | 15000 |
| Standard Deviation of of Training Samples | 6753 | 1453 |
| Averaged Number of Testing Samples | 2419 | 1666 |
| Standard Deviation of Number of Testing Samples | 742 | 165 |
| Image per Class | 10,**50** | 10,**50** |
| Local Update Epochs | **1**,2,5 | **1**,2,5 |
| Local Distillation Update Epochs | 50, **100**, 200 | 50, **100**, 200 |
| global Distillation Update Epochs | 200, **500**, 1000 | 200, **500**, 1000 |
| $\lambda$ | 1 | 1 |

Table 14: `RETINA` settings for all federated learning, including the number of training and testing examples and local update epochs. Image per class is the number of distilled images used for distribution matching only in FEDLGD. The image size is set to $96 \times 96$.

| Datasets | Drishti | Acrima | RIM | Refuge |
|---|---|---|---|---|
| Number of clients | 1 | 1 | 1 | 1 |
| Number of Training Samples | 82 | 605 | 385 | 1000 |
| Number of Testing Samples | 19 | 100 | 100 | 200 |
| Image per class | 10 | 10 | 10 | 10 |
| Local Distillation Update Epochs | 100 | 100 | 100 | 100 |
| global Distillation Update Epochs | 500 | 500 | 500 | 500 |
| $\lambda$ | 0.1 | 0.1 | 0.1 | 0.1 |

### E.6 MEMBERSHIP INFERENCE ATTACK

Studies show that neural networks are prone to suffer from several privacy attacks such as Membership Inference Attacks (MIA) (Shokri et al., 2017). In MIA, the attackers have a list of *query* data, and the purpose is to determine whether the *query* data belongs to the original training set. As discussed in (Dong et al., 2022; Xiong et al., 2022), using distilled data to train a target model can defend against multiple attacks up to a certain level. We will especially apply MIA to test whether our work can defend against privacy attacks. In detail, we perform MIA directly on models trained with FedAvg (using the original data set) and FEDLGD (using the synthetic dataset). We show the attack results in Figure 17 following the evaluation in (Carlini et al., 2022a). If the ROC curve intersects with the diagonal dashed line (representing a random membership classifier) or lies below it (indicating that membership inference performs worse than random chance), it signifies that the approach provides a stronger defense against membership inference compared to the method with a larger area under the ROC curve. It can be observed that models trained with synthetic data exhibit ROC curves that are more closely aligned with or positioned below the diagonal line, suggesting that attacking membership becomes more challenging.

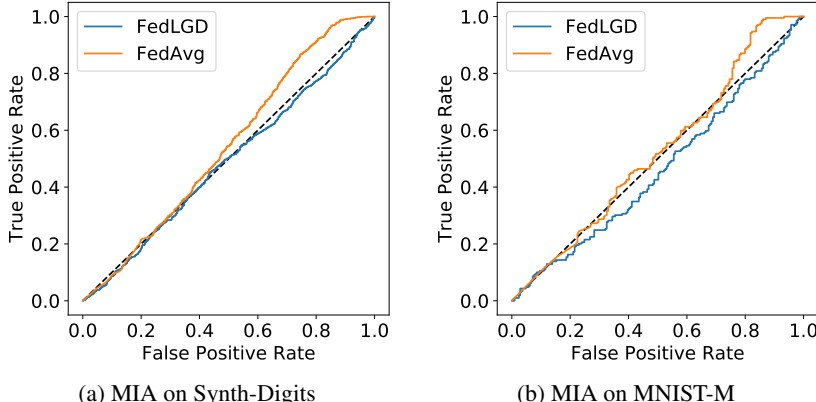

(a) MIA on Synth-Digits        (b) MIA on MNIST-M

Figure 17: MIA attack results on models trained with FedAvg (using original dataset) and FEDLGD (using distilled virtual dataset). If the ROC curve is the same as the diagonal line, it means the membership cannot be inferred. One can observe the ROC curve for the model trained with synthetic data is closer to the diagonal line, which indicates the membership information is harder to be inferred.

## F  OTHER HETEROGENEOUS FEDERATED LEARNING METHODS USED IN COMPARISON

FL trains the central model over a variety of distributed clients that contain non-iid data. We detailed each of the baseline methods we compared in Section 4 below.

**FedAvg (McMahan et al., 2017)**  The most popular aggregation strategy in modern FL, Federated Averaging (FedAvg) (McMahan et al., 2017), averages the uploaded clients' model as the updated server model. Mathematically, the aggregation is represented as $w^{t+1} = w^t - \eta \sum_{i \in S_t} \frac{|D_i|}{n} \Delta w_k^t$ (Li et al., 2021a). Because FedAVG is only capable of handling Non-IID data to a limited degree, current FL studies proposed improvements in either local training or global aggregation based on it.

**FedProx (Li et al., 2020a)**  FedProx improves local training by directly adding a $L_2$ regularization term, $\mu$, $\frac{\mu}{2}||w - w^t||^2$ controlled by hyperparameter $\mu$, in the local objection function to shorten the distance between the server and the client distance. Namely, this regularization enforces the updated model to be as close to the global optima as possible during aggregation. In our experiment, we carefully tuned $\mu$ to achieve the current results.

**FedNova (Wang et al., 2020)**    FedNova aims to tackle imbalances in the aggregation stage caused by different levels of training (e.g., a gap in local steps between different clients) before updating from different clients. The idea is to make larger local updates for clients with deep level of local training (e.g., a large local epoch). This way, FedNova scales and normalizes the clients' model before sending them to the global model. Specifically, it improves its objective from FedAvg to $w^{t+1} = w^t - \eta \frac{\sum_{i \in S_t} |D^i| \tau_i}{n} \sum_{i \in S_t} \frac{|D^i| \Delta w_k^t}{n \tau_i}$ (Li et al., 2021a).

**Scaffold (Karimireddy et al., 2020)**    Scaffold introduces variance reduction techniques to correct the 'clients drift' caused by gradient dissimilarity. Specifically, the variance on the server side is represented as $v$, and on the clients' side is represented as $v_i$. The local control variant is then added as $v_i - v + \frac{1}{\tau_i \eta}(w^t - w_i^t)$. At the same time, the Scaffold adds the drift on the client side as $w^t = w^t - \eta(\Delta(w_t; b) - v_i^t + v)$ (Li et al., 2021a).

**Virtual Homogeneous Learning (VHL) (Tang et al., 2022)**    VHL proposes to calibrate local feature learning by adding a regularization term with global anchor for local training objectives $\mathbb{E}_{(x,y) \sim P_k} l(\rho \circ \psi(x), y) + \mathbb{E}_{(x,y) \sim P_v} l(\rho \circ \psi(x), y) + \lambda \mathbb{E}_y d(P_k(\psi(x)|y), P_c(\psi(x)|y))$. They theoretically and empirically show that adding the term can improve the FL performance. In the implementation, they use untrained StyleGAN (Karras et al., 2019) to generate global anchor data and leave it unchanged during training.

A comprehensive experimental study of FL can be found here (Li et al., 2021a). Also, a survey of heterogeneous FL is here (Zhu et al., 2021a).

