# OpenReview forum: "Federated Virtual Learning on Heterogeneous Data with Local-global Distillation"
_ICLR.cc/2024/Conference — Submitted to ICLR 2024_

### Official Review · Reviewer_8zES · 2023-10-14

**Soundness:** 3 good
**Presentation:** 2 fair
**Contribution:** 3 good
**Rating:** 6
**Confidence:** 2

**Summary:**

1.	This paper proposes a federated virtual learning approach that leverages local and global dataset distillation techniques to simultaneously tackle the challenge of data heterogeneity as well as efficient training in federated learning. The authors claim that dataset distillation can exacerbate the heterogeneity among clients’ local data and propose to alleviate this issue with distribution matching.
2.	The problem addressed in this paper is novel and interesting. The adverse effect of dataset distillation in a federated learning setting is insightful. The proposed approach seems feasible and promising.
3.	In your paper, the model on clients is split into feature extractors and classification heads. This split learning-like paradigm has been widely adopted by a series of prior works [1,2,3]. Please explain the deplorability of your approach on existing methods. More elaboration on how your proposed method relates to these works would be appreciated.
4.	If I understand you correctly, FedProx is proposed by [4] rather than [5]. Do I misunderstand something?
5.	Some of the benchmark algorithms, such as FedProx [4], Scaffold [6], are somewhat outdated. In your experiments, you have used different open-sourced datasets as private data for clients, and this degree of data heterogeneity is apparently unfavorable for the regularization-based methods mentioned above. Would it be possible to compare your approach with some novel federated learning methods based on GANs [7], which seem to be more suitable for your scenario?

[1]  "FedICT: Federated Multi-task Distillation for Multi-access Edge Computing." IEEE Transactions on Parallel and Distributed Systems (2023).

[2] "Group knowledge transfer: Federated learning of large cnns at the edge." Advances in Neural Information Processing Systems 33 (2020): 14068-14080.

[3] "Exploring the distributed knowledge congruence in proxy-data-free federated distillation." arXiv preprint arXiv:2204.07028 (2022).

[4] "Federated optimization in heterogeneous networks." Proceedings of Machine learning and systems 2 (2020): 429-450.

[5] "On the convergence of fedavg on non-iid data." arXiv preprint arXiv:1907.02189 (2019).

[6] "Scaffold: Stochastic controlled averaging for federated learning." International conference on machine learning. PMLR, 2020.

[7] "Data-free knowledge distillation for heterogeneous federated learning." International conference on machine learning. PMLR, 2021.

**Strengths:**

The problem addressed in this paper is novel and interesting. The adverse effect of dataset distillation in a federated learning setting is insightful. The proposed approach seems feasible and promising.

**Weaknesses:**

1.In your paper, the model on clients is split into feature extractors and classification heads. This split learning-like paradigm has been widely adopted by a series of prior works [1,2,3]. Please explain the deplorability of your approach on existing methods. More elaboration on how your proposed method relates to these works would be appreciated.

2.If I understand you correctly, FedProx is proposed by [4] rather than [5]. Do I misunderstand something?

3.Some of the benchmark algorithms, such as FedProx [4], Scaffold [6], are somewhat outdated. In your experiments, you have used different open-sourced datasets as private data for clients, and this degree of data heterogeneity is apparently unfavorable for the regularization-based methods mentioned above. Would it be possible to compare your approach with some novel federated learning methods based on GANs [7], which seem to be more suitable for your scenario?

[1]  "FedICT: Federated Multi-task Distillation for Multi-access Edge Computing." IEEE Transactions on Parallel and Distributed Systems (2023).

[2] "Group knowledge transfer: Federated learning of large cnns at the edge." Advances in Neural Information Processing Systems 33 (2020): 14068-14080.

[3] "Exploring the distributed knowledge congruence in proxy-data-free federated distillation." arXiv preprint arXiv:2204.07028 (2022).

[4] "Federated optimization in heterogeneous networks." Proceedings of Machine learning and systems 2 (2020): 429-450.

[5] "On the convergence of fedavg on non-iid data." arXiv preprint arXiv:1907.02189 (2019).

[6] "Scaffold: Stochastic controlled averaging for federated learning." International conference on machine learning. PMLR, 2020.

[7] "Data-free knowledge distillation for heterogeneous federated learning." International conference on machine learning. PMLR, 2021.

**Questions:**

1.In your paper, the model on clients is split into feature extractors and classification heads. This split learning-like paradigm has been widely adopted by a series of prior works [1,2,3]. Please explain the deplorability of your approach on existing methods. More elaboration on how your proposed method relates to these works would be appreciated.

2.If I understand you correctly, FedProx is proposed by [4] rather than [5]. Do I misunderstand something?

3.Some of the benchmark algorithms, such as FedProx [4], Scaffold [6], are somewhat outdated. In your experiments, you have used different open-sourced datasets as private data for clients, and this degree of data heterogeneity is apparently unfavorable for the regularization-based methods mentioned above. Would it be possible to compare your approach with some novel federated learning methods based on GANs [7], which seem to be more suitable for your scenario?

[1]  "FedICT: Federated Multi-task Distillation for Multi-access Edge Computing." IEEE Transactions on Parallel and Distributed Systems (2023).

[2] "Group knowledge transfer: Federated learning of large cnns at the edge." Advances in Neural Information Processing Systems 33 (2020): 14068-14080.

[3] "Exploring the distributed knowledge congruence in proxy-data-free federated distillation." arXiv preprint arXiv:2204.07028 (2022).

[4] "Federated optimization in heterogeneous networks." Proceedings of Machine learning and systems 2 (2020): 429-450.

[5] "On the convergence of fedavg on non-iid data." arXiv preprint arXiv:1907.02189 (2019).

[6] "Scaffold: Stochastic controlled averaging for federated learning." International conference on machine learning. PMLR, 2020.

[7] "Data-free knowledge distillation for heterogeneous federated learning." International conference on machine learning. PMLR, 2021.

---

> ### Author Response · Authors · 2023-11-19
> **Discussion about the differences and deplorability with methods that also split the model into feature extractor and classification heads**
>
> Thanks for the questions. We appreciate the opportunity to point out the differences between FedLGD and the papers you suggested to highlight our novel approach and to discuss the potential deployment of FedLGD with the methods.
>
> - **We have different setups.** Although we both formulate a classification model into a feature extractor and a classification head, FedLGD always keeps forward and backward training on both parts in local clients; however, those methods require putting the classification head on the server for inference and updates.
> - **We are orthogonal approaches.** FedLGD split the model intro feature extractor and classification head so that we can add *regularization to feature space* with *different data* (local and global distilled) on the *same local feature extractor*. In contrast, those papers are knowledge distillation methods that split the model so that they can obtain and share local features with the server. Then, they perform *regularization on logits* from the *same data* feature on *different classification heads.*
> - **Those methods make assumptions in feature sharing, but FedLGD does NOT** To enable the comparison of logits for knowledge distillation, those methods require sharing features from the local feature extractor with the server. Differently, FedLGD does not require any data feature sharing and we only share gradients following classical FL during the FL model updating stage. We consider that the sharing feature and additional knowledge may violate the intention of improving privacy in federated virtual learning.
>
> Given the fundamental differences in the approaches and assumptions on information sharing, we believe our methods and those methods are not directly comparable. In fact, FedLGD and the model knowledge distillation-based methods can be combined as orthogonal strategies to address heterogeneity issues in FL. For example, the distilled global virtual data can serve as an ideal candidate to perform knowledge distillation as it roughly captures global data distribution. Thus, it can eliminate the requirement of sharing real data features or using public datasets. We hope our approach can shed light on leveraging virtual data in FL and inspire future work on the interesting exploration of combining our strategy with others.
>
> We thanks the reviewer for pointing us to the interesting papers, and we have added them in our related work.
>
> [1] "FedICT: Federated Multi-task Distillation for Multi-access Edge Computing." IEEE Transactions on Parallel and Distributed Systems (2023).
>
> [2] "Group knowledge transfer: Federated learning of large cnns at the edge." Advances in Neural Information Processing Systems 33 (2020): 14068-14080.
>
> [3] "Exploring the distributed knowledge congruence in proxy-data-free federated distillation." arXiv preprint arXiv:2204.07028 (2022).

---

> > ### Comment · Reviewer_8zES · 2023-11-19
> >
> > Thank you for your careful rebuttal. I am impressed by your idea of using distilled global virtual data as the candidate for knowledge distillation. Personally, I suggest clarifying the difference between knowledge distillation and dataset distillation in your paper, as the former technique combined with federated learning has accumulated rich literature and a large number of readers. Anyway, I have increased my rating in my revised official comments.

---

> ### Author Response · Authors · 2023-11-19
> **Apologize for the wrong citation**
>
> We thank the reviewer for the careful review and for pointing out this subtle error. We incorrectly referred FedProx to [2] in our Section 4.1 and 5 due to the similarity of the reference codes used by [1] and [2] (i.e., li2020a and li2020b). It should be [1] instead. We really appreciate this comment, and we have corrected it in our revision.
>
> [1] "Federated optimization in heterogeneous networks." Proceedings of Machine learning and systems 2 (2020): 429-450.
>
> [2] "On the convergence of fedavg on non-iid data." arXiv preprint arXiv:1907.02189 (2019).

---

> ### Author Response · Authors · 2023-11-19
> **Explanation of the selection of our baseline methods**
>
> We thank the reviewer for pointing out the interesting FL work, FedGen [2]. We did not include FedGen (and other knowledge distillation-based methods) in our comparison works is that we consider knowledge distillation and data distillation two orthogonal directions to solve data heterogeneity issues.
>
> It is worth noting the training strategy of federated virtual learning is quite novel and under-explored. Thus most of the existing FL methods are not suitable for making direct comparisons. In fact, we included VHL [1], which shares most similar thoughts as FedLGD by using virtual data as regularization in local training. We show that using the virtual data generated by federated gradient matching can better handle the heterogeneous data scenario.
>
> Following the suggestion, we added the work to the related work in our revision and justified the orthogonality between knowledge distillation-based FL methods and FedLGD. To address the reviewer’s question, we report the results on FedGen [2] on DIGITS, CIFAR10C(client ratio=1), and RETINA in the following table. We also include FedGen in the related work in our revision.
> .
>
> | Avg. Acc.       | FedGen| VHL     | FedLGD |
> |-------------------|------------|-----------|---------------|
> | DIGITS          | 66.9      | 86.5      | **86.7**    |
> | CIFAR10C     | 39.6      | 55.2      | **57.4**    |
> | RETINA         | 82.1      | 78.6      | **85.1**    |
>
> Observe that FedLGD consistently outperforms other methods. We conjecture the reason FedGen is not performing well for two reasons: It requires a larger number of training data to obtain a more reasonable global GAN. 2. It requires a larger feature space to better regularize local training with knowledge distillation (as shown in RETINA experiments, where it can use a large feature space, thus resulting in better performance).
>
> [1] Tang Z, Zhang Y, Shi S, He X, Han B, Chu X. Virtual homogeneity learning: Defending against data heterogeneity in federated learning. InInternational Conference on Machine Learning 2022 Jun 28 (pp. 21111-21132). PMLR.
>
> [2] Zhu Z, Hong J, Zhou J. Data-free knowledge distillation for heterogeneous federated learning. InInternational conference on machine learning 2021 Jul 1 (pp. 12878-12889). PMLR.

---

> ### Author Response · Authors · 2023-11-20
> **Thank you for your feedback!**
>
> Dear Reviewer 8zES,
>
> Thank you for the recognition of our work and rebuttal. Your comments are valuable for us to refine our manuscript. We will highlight the differences between knowledge distillation and dataset distillation in our related work session in our revision.
>
> Best Regards,
>
> FedLGD authors

---

### Official Review · Reviewer_PXsJ · 2023-10-31

**Soundness:** 3 good
**Presentation:** 2 fair
**Contribution:** 3 good
**Rating:** 5
**Confidence:** 4

**Summary:**

This paper proposes a federated learning method called FedLGD that uses local and global dataset distillation to handle data heterogeneity.FedLGD uses an iterative distillation process to generate local and global virtual datasets that mitigate data heterogeneity and improve efficiency in federated learning. The local-global distillation and feature regularization are key components that help FedLGD achieve strong performance.

**Strengths:**

1. The discover of using dataset distillation can amplify the statistical distances is interesting.
  2. Achieves state-of-the-art results on benchmark datasets with domain shifts, outperforming existing federated learning algorithms.

**Weaknesses:**

1. t-SNE figures are not represented as vectors.

2. Sharing gradients from clients to the server for a global virtual data update may pose security risks. Some attacks could potentially reconstruct raw data using gradient information, similar to the risks associated with Deep Gradient Leakage. Why sharing averaged gradients is safe?

3. What is the rationale behind clients requiring local virtual data instead of training directly on their local private data?

4. Could you clarify why this method has not been compared to other FL methods utilizing dataset distillation?

**Questions:**

See Weaknesses.

---

> ### Author Response · Authors · 2023-11-19
> **Justification and explanation of t-SNE plot**
>
> We thank the reviewer for the clarification comment. We chose to show the 2-D plot for more intuitive visualization. In our original submission, we also reported the **statistical distances** between the two example datasets from two clients to support the heterogeneity statement. Specifically, in **the second paragraph of our Introduction**  we stated “Quantitatively, we found that using dataset distillation can amplify the statistical distances between the two datasets, with Wasswestein Distance and Maximum Mean Discrepancy (MMD) (Gretton et al., 2012) both increasing by around 40%.” We believe that the statistical distance and t-SNE plots can provide a reasonable evidence to our finding.

---

> ### Author Response · Authors · 2023-11-19
> **Justification of the privacy preservation in FedLGD**
>
> We thank the reviewer for the constructive comment. Yes, privacy is indeed an important concern in federated learning. Thus, we discuss the privacy guarantee of FedLGD in the **last paragraph in Section 3** and we highlight the statements as follows:
>
> - FedLGD does not share additional client information compared to baseline methods such as FedAvg [1].
> - Notably, the gradients we share in FedLGD are w.r.t. Local virtual data distilled by Distribution Matching [2], which has been shown to defend against Membership Inference Attack and Reconstruction Attack [3]. We also show our empirical defense results against Membership Inference Attack in Appendix E.6.
> - Privacy preservation can be further improved by employing differential privacy [4] in dataset distillation, but this is beyond the main focus of our work.
>
> [1] McMahan B, Moore E, Ramage D, Hampson S, y Arcas BA. Communication-efficient learning of deep networks from decentralized data. InArtificial intelligence and statistics 2017 Apr 10 (pp. 1273-1282). PMLR.
>
> [2] Zhao B, Bilen H. Dataset condensation with distribution matching. InProceedings of the IEEE/CVF Winter Conference on Applications of Computer Vision 2023 (pp. 6514-6523).
>
> [3] Dong T, Zhao B, Lyu L. Privacy for free: How does dataset condensation help privacy?. InInternational Conference on Machine Learning 2022 Jun 28 (pp. 5378-5396). PMLR.
>
> [4] Abadi M, Chu A, Goodfellow I, McMahan HB, Mironov I, Talwar K, Zhang L. Deep learning with differential privacy. InProceedings of the 2016 ACM SIGSAC conference on computer and communications security 2016 Oct 24 (pp. 308-318).

---

> ### Author Response · Authors · 2023-11-19
> **Rationale behind using local virtual data instead of local raw data for FL**
>
> We thank the reviewer for the clarification question. As we stated **in our abstract**, training with distilled virtual data can provide higher training efficiency and alleviate the problem of inference attacks, as demonstrated in centralized settings [1].
>
> Building on this, we explore its usage in FL (named federated virtual learning - FVL), because privacy and efficiency (computation cost) are common concerns in FL, together with its obvious assistance in synchronization.
>
> Specifically, the benefits are:
>
> - **Synchronization**: Training with a small number of representative virtual data can facilitate local training, which can improve FL synchronization.
> - **Computation Cost**: As shown in Appendix D.3 and Table 6, we empirically show that training with virtual data requires a much lighter computation cost. (We’ve corrected the typo in Appendix D.3 in our revision.)
> - **Privacy**: As reported in [1], training with virtual data can protect the model from privacy attacks such as Inversion Attack [2] and Membership Inference Attack (MIA) [3]. In addition, as stated in **the last paragraph of Section 3**, “we consider averaged gradients w.r.t. local virtual data and the method potentially defends inference attacks better (Appendix E.6).”
>
> The core of this work aims to resolve the heterogeneity issue associated with distillation-based virtual data in FVL. This will make the use of virtual data more feasible in FL and lead to improved synchronization, efficiency, and privacy while approaching the utility of real data.
>
> [1] Dong T, Zhao B, Lyu L. Privacy for free: How does dataset condensation help privacy?. InInternational Conference on Machine Learning 2022 Jun 28 (pp. 5378-5396). PMLR.
>
> [2] Geiping J, Bauermeister H, Dröge H, Moeller M. Inverting gradients-how easy is it to break privacy in federated learning?. Advances in Neural Information Processing Systems. 2020;33:16937-47.
>
> [3] Shokri R, Stronati M, Song C, Shmatikov V. Membership inference attacks against machine learning models. In2017 IEEE symposium on security and privacy (SP) 2017 May 22 (pp. 3-18). IEEE.

---

> ### Author Response · Authors · 2023-11-19
> **Explanation of not comparing to other FL+data distillation works**
>
> We thank the reviewer for raising this question. As we stated in the related work (Sec. 2.2), to our best knowledge, the current FL+Data distillation works typically share local distilled data to server [1,2,3], which we consider a non-practical and more privacy-sensitive operation. Furthermore, some methods (e.g., [1]) need to use real data as initialization to obtain good performance.
>
> On the contrary, due to privacy concerns, we choose to train the model locally and share the gradients w.r.t. Local virtual data,  which is the default operation in classical FL, such as FedAvg. Therefore, the other FL+data distillation settings are significantly different from ours regarding the information shared and data leakage. We discussed these studies in our related work but believed it was unfair to make a direct comparison, given their special request for information sharing.
>
> [1] Xiong Y, Wang R, Cheng M, Yu F, Hsieh CJ. Feddm: Iterative distribution matching for communication-efficient federated learning. In Proceedings of the IEEE/CVF Conference on Computer Vision and Pattern Recognition 2023 (pp. 16323-16332).
>
> [2] Goetz J, Tewari A. Federated learning via synthetic data. arXiv preprint arXiv:2008.04489. 2020 Aug 11.
>
> [3] Hu S, Goetz J, Malik K, Zhan H, Liu Z, Liu Y. FedSynth: Gradient Compression via Synthetic Data in Federated Learning. InWorkshop on Federated Learning: Recent Advances and New Challenges (in Conjunction with NeurIPS 2022) 2022 Oct 21.

---

> ### Author Response · Authors · 2023-11-22
> **Thank you for your valuable comments and we sincerely look forward to your feedback for rebuttal**
>
> Dear Reviewer PXsJ,
>
> As the rebuttal deadline is approaching, we would like to know if our rebuttals have addressed your concerns and questions. Also, we have tried our best to reflect your brilliant feedback to our revision. We appreciate the opportunity to discussing during this stage and are delighted to address your further question if there is any. If you are satisfied with our response and revision, we would be grateful if you could kindly re-consider the rating for FedLGD.
>
> Best Regards,
>
> FedLGD authors

---

### Author Response · Authors · 2023-11-19
**Overall Response**

We thank the reviewers for their valuable time and efforts in reviewing FedLGD, which we summarize as follows:

**Strength**

- Our discovery that data distillation can amplify statistical distance is interesting and insightful (PXsJ, 8zES)
- The experimental results on domain-shift benchmark datasets show that FedLGD outperforms existing federated learning algorithms(PXsJ), which makes FedLGD seem feasible and promising(8zES).
- The problem addressed in FedLGD is novel and interesting.(8zES)

**Motivation**

We would like to take a chance to re-state the motivation of FedLGD to clarify the confusion from reviewer PXsJ.

> Explanation of using local virtual data instead of local raw data for FL

The overall motivation of FedLGD starts from the advantages of using distilled virtual data for (centralized) model training: efficiency and privacy protection against inversion and membership inference attacks. We further explore its usage in federated learning (named federated virtual learning - FVL), as it further improves the synchronization in federated learning in addition to the aforementioned advantages. However, we noticed that there will be amplified heterogeneity issues, which motivates the design of FedLGD to facilitate FVL.

**Clarification**

> Differences between FedLGD and other FL + data distillation method

Reviewer PXsJ asks why we don’t compare with existing FL+data distillation papers.
We clarify that existing FL+data distillation papers usually share local distilled data to the server, which we considered a dangerous move due to privacy concerns. Instead, we consider using a classic FL pipeline with virtual data - training local virtual data on the clients’ side and aggregating the gradients on the server’s side (named federated virtual learning - FVL)

> Differences between FedLGD and other FL methods

- Reviewer 8zES suggests comparing with more recent methods and comparing with FedGen [1], a knowledge distillation-based method. We clarify that FVL is a relatively under-explored idea and we chose the most related work, including VHL [2]. We also explain that other knowledge distillation-based methods and our data distillation strategy are two orthogonal directions to solve data heterogeneity issues. Still, we perform experiments following the reviewer’s suggestion, and the results show that FedLGD can outperform FedGen in our data heterogeneous scenario.

- Reviewer 8zES asks about the deplorability of FedLGD with [3,4,5] that also split the feature extractor and classification head.
We state the fundamental differences between FedLGD and the methods based on the significantly different model updating strategies and the differences in shared local information - [3,4,5] share data features to server and perform server model update, but FedLGD does NOT. Then, we provide a plausible discussion on how to deploy the data distillation idea of FedLGD with the listed methods.

[1] Zhu Z, Hong J, Zhou J. Data-free knowledge distillation for heterogeneous federated learning. InInternational conference on machine learning 2021 Jul 1 (pp. 12878-12889). PMLR.

[2] Tang Z, Zhang Y, Shi S, He X, Han B, Chu X. Virtual homogeneity learning: Defending against data heterogeneity in federated learning. InInternational Conference on Machine Learning 2022 Jun 28 (pp. 21111-21132). PMLR.

[3] "FedICT: Federated Multi-task Distillation for Multi-access Edge Computing." IEEE Transactions on Parallel and Distributed Systems (2023).

[4] "Group knowledge transfer: Federated learning of large cnns at the edge." Advances in Neural Information Processing Systems 33 (2020): 14068-14080.

[5] "Exploring the distributed knowledge congruence in proxy-data-free federated distillation." arXiv preprint arXiv:2204.07028 (2022).

**Updates on the revision**

- Add the papers discussed by the reviewers to the related work section and highlight the orthogonality of other knowledge distillation methods and ours
- Carefully corrected minor typos.


We have carefully addressed the reviewers’ comments and have updated the manuscript accordingly. The comments helped us revise our paper to a better shape. We kindly request the reviewers to evaluate our responses and the manuscript, and we are more than happy to answer any further questions that may arise.

Best Regards,

FedLGD authors

---

### Meta-Review · Area_Chair_K1ma · 2023-12-20

**Metareview:**

In this paper the authors focused on the problem federated learning and propose an interesting framework centered on distilling data -- on each local worker, it uses feature matching to generate virtual data (use the global feature extractor to improve the quality) and on the global worker, it uses gradient matching to generate virtual data.

Reviewers found this paper to be interesting, however, has several concerns -- (1) the precise privacy guarantee that this method provides and more comprehensive study on its benefit against previous related work; (2) comparisons with other FL methods, especially utilizing dataset distillation.

With all these considered, the reviewers are leaning towards rejection and hope that the authors can integrate these feedback for future submissions.

**Justification For Why Not Higher Score:**

It would be great if this paper can provide more studies on the privacy guarantee/benefit that the proposed method provides, and more comprehensively compare with other methods.

**Justification For Why Not Lower Score:**

N/A

---

### Decision · Program_Chairs · 2024-01-16

Reject